# Structured Visual Landscape: Generating Preferred Representations in Multi-modal Biological and Artificial Neural Networks

## Abstract

Understanding how neurons responding to visual stimulus inputs is an important question in both deep learning and neuroscience. It has significant implications in enhancing the interpretability of black-box artificial neural networks and understanding the visual representation in biological neural networks. We proposed a structured visual representation landscape and design an activation score based prior that allows effectively regularizing the landscape with either activations from a brain region or units in neural networks. Our model **Vis-Lens** integrates a variational auto-encoder and diffusion model as an image generative model. It allows generation of natural realistic preferred images with directly modifying the activation-regularized latents, which avoids the tedious optimization procedure. We demonstrate the effectiveness of our framework in both artificial neural networks and biological neural networks with multi-modal response data derived from human visual cortex, including functional Magnetic Resonance Imaging (fMRI) and electroencephalography (EEG). Our framework outperforming state-of-the-art method on generating visual representations of those networks.

## 1 Introduction

The exploration of visual representations in both biological and artificial neural networks has advanced our understanding in how complex visual stimuli are processed and represented in neural networks (Marr, 2010; Hubel & Wiesel, 1962; Bashivan et al., 2019). Previous studies that demonstrate the capacity of neurons in the medial temporal lobe (MTL) of the human brain to form invariant representations of complex stimuli (Quiroga et al., 2005), such as faces, landmarks, and objects, regardless of visual variations. These findings illustrate how single neurons can encode high-level, abstract percepts with remarkable specificity, leading to the hypothesis of sparse and invariant coding mechanisms in the brain (Olshausen & Field, 2004).

While methods exploring visual representation in some brain regions work fine by learning directly from embeddings of popular image encoders like CLIP (Garcia Cerdas et al., 2025; Luo et al., 2023b), understanding the feature representations for many brain regions that are not well-studied remains challenging. For example, some higher-order brain regions can demonstrate mix-selectivity (Rigotti et al., 2013) where they can respond to images with small overlap in shared features. As presented in Figure 1, the top images that maximally activate the brain region distributed across the embeddings space of CLIP encoder (Radford et al., 2021a).

Additionally, brain activity measured is often noisy, low-resolution, and partially observed, which introduces the difficulties for analysis and modeling. Unlike previous works that aim to reconstruct entire images from brain activities (Naselaris et al., 2011; Nishimoto et al., 2011; Shen et al., 2019; Horikawa & Kamitani, 2017), our work focuses on understanding the feature representation of brain regions by generating new preferred images that maxmizing or minizing the activity of those regions. Those synthetic visual stimuli can be deployed in follow-up neuroscience experiments to test new hypotheses, refine ROI functional maps, and accelerate discovery by guiding stimulus design.

To achieve this goal, we proposed Vis-Lens, an effective approach to interpret the feature represented by brain regions. Given the noisy nature of brain recordings, instead of using brain activities directly as input to the decoding model, our method refines the landscape of visual representation by em-

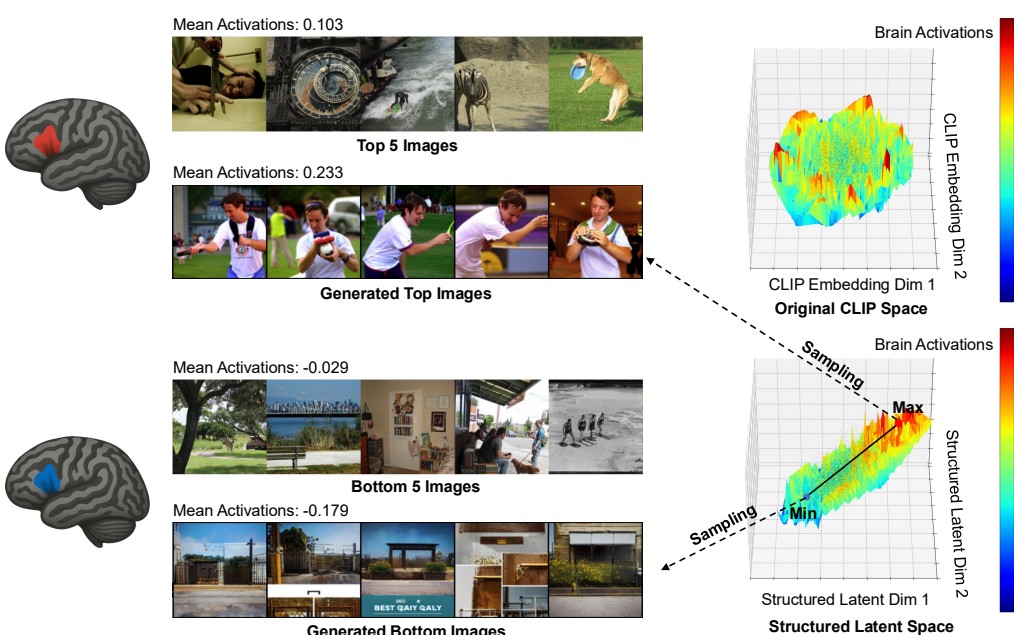

Figure 1: **Original CLIP Space (Top):** Original CLIP embedding space without any activation constraint: image embeddings that elicit high and low neural responses are dispersed and not clustered throughout the space without shared visual structure. **Structured Landscape (Bottom):** By encoding CLIP embeddings into the regularized latent space constrained with neuronal activations as the *prior*, our method Vis-Lens could provide a more structured and representative landscape to better identify the representations and preferences of brain regions than sorting top-k images in dataset. The results are based on brain activations from the NSD dataset (Allen et al., 2021). In the landscape plots, the color bar and z axis indicate the activation of brain region given different images as inputs, x and y axis represent two latent dimensions, respectively.

ploying brain activation to regularize the prior, serving as a soft constraint. We further integrate the structured landscape with powerful variational auto-encoder (VAE) and diffusion generative models to generate realistic preferred images that can effectively modulate the brain activations.

Our method can facilitate the exploration of visual representations of brain regions while maintaining interpretability. It is validated with multi-modal data, including fMRI data from the Natural Scenes Dataset (NSD) (Allen et al., 2021) and EEG data from THINGS-EEG dataset (Grootswagers et al., 2022; Gifford et al., 2022), analyzing responses in the human visual cortex to complex natural stimuli. Meanwhile, due to the scarcity of image response data from human brains, our proposed framework is also extended to artificial neural networks (ANNs), allowing controlled testing of visual feature representations and interpretability in artificial systems. By integrating insights from both biological and artificial neural networks, we aim to bridge understanding of visual representation strategies across natural and engineered systems. Our contributions are outlined as follows:

- We introduce an activation-regularized prior for VAE (Kingma & Welling, 2022) that organizes its latent space into two distinct clusters, corresponding to codes that increase or decrease a target brain region's activation. This design enables easy plug-and-play integration with conventional generative models.

- Our method synthesizes visual representations from specified brain regions, outperforming the state-of-the-art in both activation change and perceptual realism. Moreover, it provides a generalized framework showing robustness on multi-modal and cross-subject biological data, and can also transfer to visualizing features in artificial neural networks.

- By generating controllable, high-quality visual stimuli that can modulate specific regions, our framework demonstrates new potentials in vision-based brain computer interface applications and can be used to generate new hypothesis for future neuroscience studies.

## 2 RELATED WORKS

**Image generative models.** Advances in generative models have significantly transformed image synthesis. Foundational approaches such as GANs (Goodfellow et al., 2014) led to extensions like Wasserstein GANs (Arjovsky et al., 2017) and Conditional GANs (Mirza & Osindero, 2014), with fidelity improvements using spectral normalization (Miyato et al., 2018), while VAEs (Kingma & Welling, 2013) introduced probabilistic encoding, later combined with latent space regularization (Ho et al., 2022; Van den Oord et al., 2016). Further, Cascaded generation (Ho et al., 2022) and autoregressive modeling (Chen et al., 2018) enhanced synthesis quality. More superior generation models like Diffusion models (Ho et al., 2020; Rombach et al., 2022) advanced controllable, high-fidelity synthesis, integrating structured perturbations via SDEdit (Meng et al., 2022) and text-to-image alignment through CLIP (Radford et al., 2021b). IP-Adapter (Ye et al., 2023) refined latent spaces for semantic consistency, while GLIDE (Nichol et al., 2022) and Stable Diffusion (Saharia et al., 2022) further demonstrated photorealistic, text-guided image generation. Our work is developed based on these conventional generative model frameworks, which adopts VAEs to build up our proposed structured landscape and a Stable Diffusion model generation pipeline to synthesize preferred image conditioned on CLIP embeddings derived from the structured landscape.

**Feature visualization.** Visualization techniques provide deep insights into neural network interpretability and functionality, spanning both biological and artificial systems. Early work by Erhan *et al.* (Erhan et al., 2009) focused on minimal regularization approaches, while adversarial examples exposed vulnerabilities in neural networks (Szegedy et al., 2013). Mahendran and Vedaldi (Mahendran & Vedaldi, 2015) and Nguyen *et al.* (Nguyen et al., 2015) further expanded on interpretability with total variation regularization, counterexamples, and image blurring. Meanwhile, techniques such as DeepDream by Mordvintsev *et al.* (Mordvintsev et al., 2015; 2016), employing jitter, multi-scale visualization, and gradient normalization (Ø ygard, 2015; Tyka, 2016), enhanced clarity in feature visualization. For prior related work, Nguyen *et al.* (Nguyen et al., 2016) demonstrated generative adversarial synthesis of preferred inputs, emphasizing controlled image generation through learned priors. Our work extends this line by generating images with preferred features based on pre-defined priors in landscape shaped by activations.

**Brain decoding and most exciting images.** Generative models leveraging brain activity as direct input have substantially advanced neural-to-visual synthesis. Direct neural-to-image synthesis methods convert fMRI activity into high-fidelity pictures with diffusion or GAN backbones—for example BrainDiffusion and NeuroDM (Luo et al., 2023a; Qian et al., 2024), Brain2GAN (Dado et al., 2024), latent-diffusion variants (Ferrante et al., 2024; Ozcelik & VanRullen, 2023), and energy-guided approaches (Pierzchlewicz et al., 2023), while activation-optimized methods like Inception Loops (Walker et al., 2019) and NeuroGen (Gu et al., 2022) reveal neuronal tuning by iterating stimuli based on neural responses. Recent multi-modal approaches (Huang et al., 2021b;a; Ding et al., 2023; van Gerven, 2021; Qiu et al., 2025; Benara et al., 2024) integrate generative models for improved synthesis precision, while Controllable Mind Visual Diffusion (Zeng et al., 2023), Seeing Beyond the Brain (Chen et al., 2023), and Reconstructing the Mind's Eye (Scotti et al., 2023) leverage structured diffusion priors for enhanced visual reconstructions. Though dominated by fMRI data, (Song et al., 2024; Li et al., 2024) explored this task on EEG data, obtained promising results.

To further modulate activations, computational models (Papale et al., 2024; Murty et al., 2021) and frameworks targeting single neuron contributions (Bau et al., 2017; Olah et al., 2017; Ritter et al., 2017) provide interpretable mappings of cortical activations to specific visual features. Latent space optimization strategies (Robinson et al., 2023; Xia et al., 2024) extend foundational decoding frameworks (Naselaris et al., 2011; Nishimoto et al., 2011; Shen et al., 2019; Horikawa & Kamitani, 2017) to achieve targeted neural activation modulation. BrainDiVE and BrainSCUBA(Luo et al., 2023b; 2024) uses a CLIP-based encoder and diffusion models to improve stimulus quality and semantic specificity. BrainACTIV (Garcia Cerdas et al., 2025), as the recent state-of-the-art and the first work on using diffusion models to generating images to regulate brain activations, extends these paradigms by conditioning synthesis with linearly fitting CLIP embeddings with brain activation pattern. Though modeling activations via CLIP embeddings does work in previous work, our method tries to take the noisy and mix-selective nature of brain activity data into consideration, building a novel structured landscape in VAE through CLIP embeddings, permitting more controllable and effective activation manipulation, and avoiding falling into local minimum in the optimization process.

Figure 2: **Overview of Vis-Lens**. The illustration depicts a two-stage approach integrating **a structured landscape and variational autoencoder (VAE) generation** with CLIP encoder and diffusion based generative model. **Phase I:** The input image, $i_{\text{train}}$, is encoded by a pretrained CLIP encoder $\mathcal{E}$, CLIP embeddings $x$ are encoded into a latent representation $z$ by the VAE encoder. A regularization step using a **activation-based prior** $\mathcal{N}(\mu_{\text{prior}}, \frac{1}{|r|})$ modifies the latent space for better alignment with activations, producing a regularized latent $z$. The VAE is also trained to reconstruct the input CLIP embeddings $x$ with $x_{recon}$. **Phase II:** The test images $i_{\text{test}}$, is encoded by same CLIP encoder $\mathcal{E}$. Then, with fully trained VAE and the structured visual latent representation space, we can directly **modify latents** constrained by a specific feature represented by the selected regions, obtaining a new latent $z'$. This transformed latent is decoded back into an intermediate generated CLIP embeddings $x_{gen}$, which is used to guild the image generation through a diffusion pipeline, yielding the final output $i_{\text{gen}}$ as preferred images to modulate the activation $r$.

## 3 METHODOLOGY

### 3.1 PROBLEM SETTING AND NOTATION

Let $\mathcal{I}$ be the space of natural images, $\mathcal{X}$ be the space of CLIP embeddings, and $\mathcal{Z}$ the latent space of a variational auto-encoder (VAE). For an input image $i \in \mathcal{I}$, the pretrained CLIP encoder $\mathcal{E}$ maps $i$ to a CLIP embedding $x = \mathcal{E}(i) \in \mathcal{X}$. Subsequently, the VAE encoder $q_\phi(z \,|\, x)$ produces a latent $z \in \mathcal{Z}$ with mean $\mu$ and diagonal covariance $\Sigma = \text{diag}(\sigma^2)$; the VAE decoder $p_\theta(x \,|\, z)$ reconstructs an embedding $x_{\text{rec}}$. Throughout, $r$ denotes the (scalar) activation of a chosen region of interest (ROI) in human brain or neural networks, and $\lambda$, $\omega_{\text{rec}}$, $\omega_{\text{KLD}}$ are scalar hyper-parameters.

### 3.2 CONSTRUCT STRUCTURED VAE LANDSCAPE WITH NEURONAL ACTIVATION AS PRIOR

To achieve explainable and controllable visual representation generation of a specific region or unit given the corresponding activations, we integrate the activation $r$ into the VAE prior and then construct a structured latent space in VAE. The details are described as follows:

The KL divergence (KLD) term in the VAE measures the divergence between the encoder's approximate posterior, $q_\phi(\mathbf{z}|\mathbf{x})$, and the prior, $p_\theta(\mathbf{z})$. We begin with its standard definition:

$$D_{KL}(q_\phi(\mathbf{z}|\mathbf{x}) \,||\, p_\theta(\mathbf{z})) = \int q_\phi(\mathbf{z}|\mathbf{x}) \log \frac{q_\phi(\mathbf{z}|\mathbf{x})}{p_\theta(\mathbf{z})} d\mathbf{z} = \mathbb{E}_{\mathbf{z} \sim q_\phi(\mathbf{z}|\mathbf{x})} [\log q_\phi(\mathbf{z}|\mathbf{x}) - \log p_\theta(\mathbf{z})] \quad (1)$$

Both the posterior and the prior are defined as multivariate Gaussian distributions with diagonal covariance matrices, In our framework, we have the approximate posterior: $q_\phi(\mathbf{z}|\mathbf{x}) = \mathcal{N}(\mathbf{z}; \boldsymbol{\mu}, \text{diag}(\boldsymbol{\sigma}^2))$. For the prior, $p_\theta(\mathbf{z})$, we designed a novel **activation constrained prior**: $p_\theta(\mathbf{z}) = \mathcal{N}(\mathbf{z}; \boldsymbol{\mu}^{\text{prior}}, \text{diag}((\boldsymbol{\sigma}^{\text{prior}})^2))$, which depends on the activation value $r$:

$$\sigma^{\text{prior}} = \left| \frac{\lambda}{r + \epsilon} \right|, \qquad \mu_i^{\text{prior}} = \begin{cases} \mu_{\text{pos}}, & \text{if } r > 0, \\ \mu_{\text{neg}}, & \text{otherwise,} \end{cases} \tag{2}$$

where $\lambda$ is a scaling hyperparameter, $\epsilon$ is a small constant to prevent division by zero, and $\mu_{\text{pos}}, \mu_{\text{neg}}$ are predefined means for positive and negative activations, respectively.

Given the definition of approximate posterior and activation constrained prior, the expectation in Eq 1 can be computed analytically, yielding the closed-form expression for the KLD between two distributions. Also, we assume that VAE latent space has $d_z$ dimensions, and the overall divergence is calculated across the dimensions, so the final KLD affected by activations can be written as:

$$\text{KLD}_{\text{activation}} = \sum_{i=1}^{d_z} \left[ \log \frac{\sigma_i^{\text{prior}}}{\sigma_i} - \frac{1}{2} + \frac{\sigma_i^2 + (\mu_i - \mu_i^{\text{prior}})^2}{2(\sigma_i^{\text{prior}})^2} \right] \tag{3}$$

where $\mu_i$ and $\sigma_i$ are the mean and standard deviation of the approximate posterior for the $i$-th latent dimension, respectively; $\mu_i^{\text{prior}}$ and $\sigma_i^{\text{prior}}$ are the mean and standard deviation of the prior distribution for the $i$-th dimension.

With this activation constrained prior, we can build a structured VAE latent space. Intuitively, when $r$ approximates infinite, the constrain would be extremely tight around the center of prior distribution.

### 3.3 Generate Preferred Visual Representations with Modified VAE

To produce preferred images, our method is divided into two phases: a training phase on previous VAE to construct the structured latent space and a generation phase to generate images. The VAE is the only trainable part in the pipeline, while other modules including pre-trained CLIP encoder ($\mathcal{E}$) and image-to-image diffusion model—remain frozen.

**Training.** During the training, only the VAE encoder $q_\phi(z|x)$ and decoder $p_\theta(x|z)$ that maps the embedding to a latent representation $z$ and reconstructs it back to $x_{\text{recon}}$ are optimized by Eq 4.

$$\mathcal{L}_{\text{VAE}} = \omega_{\text{recon}} \cdot \mathcal{L}_{\text{recon}} + \omega_{\text{KLD}} \cdot \text{KLD}_{\text{activation}}, \tag{4}$$

where $\mathcal{L}_{\text{recon}} = \|x - x_{\text{recon}}\|_2^2$ is the reconstruction loss between the original and decoded CLIP embeddings. The terms $\omega_{\text{recon}}$ and $\omega_{\text{KLD}}$ are scalar weights that balance the reconstruction fidelity against the activation-based regularization imposed by our modified KLD term in Eq 3.

**Generation.** To synthesize new images, the generation phase begins by taking an image $i$ and encoding it with frozen CLIP encoder to get CLIP embedding $x = \mathcal{E}(i)$, then, we feed $x$ into the trained VAE encoder to get the latent code $z = q_\phi(x)$ in the structured landscape we built. $z$ is then moved across the landscape towards the pre-defined positive or negative priors. The new, modified latent code $z'$ is then given to the VAE decoder, which translates it into a new, modified CLIP embedding $x' = p_\theta(x|z')$. This new embedding, now imbued with the desired activation-guiding properties, is finally passed to a frozen diffusion model to synthesize the final, preferred image $i'$.

## 4 Experiments

### 4.1 Datasets

**Neuroimaging Datasets.** To test our framework on multi-modal neuronal data, we utilize two large-scale human neuroimaging datasets. The Natural Scenes Dataset (NSD) (Allen et al., 2021) provides high-resolution fMRI recordings from eight subjects viewing thousands of MS COCO images (Lin et al., 2015). The THINGS-EEG dataset offers comprehensive EEG recordings from ten subjects

viewing thousands of images. (Grootswagers et al., 2022; Gifford et al., 2022). The response data was extracted from ROI masks in NSD and event-related potentials (ERPs) across occipital channels in THINGS-EEG, then averaged to reduce noise. They both serve as essential constraints in our framework to align the VAE latent space with observed brain responses in 4.3.1.

**Image Datasets.** To evaluate our method on artificial neural networks, we used two common image datasets. ImageNet-mini, a diverse subset of the full ImageNet dataset (Deng et al., 2009), serves as input for large-scale SimCLR (Chen et al., 2020) and ViT (Dosovitskiy et al., 2021) models to obtain unit activations in Section 4.3.2. Additionally, we use the CIFAR-10 dataset (Krizhevsky & Hinton, 2009), comprising 60,000 low-resolution images across ten classes, to validate performance of our method within a smaller three-layer CNN in Section 4.3.3.

## 4.2 MODELS

**Implementation of our framework.** Our VAE encoder employs a linear layer to map the CLIP embeddings to a latent space constrained by activations. The decoder also uses a single linear layer that reconstructs CLIP embeddings from the latent space, enabling controlled image synthesis through targeted latent manipulations. For other models, we use a pre-trained CLIP-ViT-H-14 model as our CLIP encoder(Radford et al., 2021a) that extracts 1024-dimensional embeddings for each input image. We choose Stable Diffusion v1.5 as our image generation model incorporating an IP-Adapter(Ye et al., 2023) that effectively modulates the generation process by injecting the edited CLIP embeddings, ensuring the semantic and structural consistency of the synthesized outputs.

**Baseline method.** For every ROI we re-implement the method in *BrainACTIV* (Garcia Cerdas et al., 2025), a ridge-regularised *linear* model on $\ell_2$-*normalized* CLIP embeddings, $\left[ \frac{x}{\|x\|} \cdot \mathbf{w} + b \right] \implies r$, yielding a single modulation vector $\mathbf{w} \in \mathbb{R}^{1024}$. This vector is interpreted as the direction of maximal and minimal activation in CLIP space: $\mathbf{z}_{\max} = \mathbf{w}/\|\mathbf{w}\|$ and $\mathbf{z}_{\min} = -\mathbf{w}/\|\mathbf{w}\|$. At inference time, a test CLIP embedding is shifted *only* along $\mathbf{z}_{\max}$ or $\mathbf{z}_{\min}$ before being injected—via an unchanged adapter—into a frozen image-to-image diffusion model.

**In-silico brain simulator.** In our experiments, brain activations are evaluated with a *DINO-ViT* encoder (Oquab et al., 2024): DINO acts as a frozen feature extractor whose 12 layer outputs are fed to a single-layer ViT ensemble trained with ridge loss to predict neuronal responses. In line with the findings from Garcia Cerdas et al. (2025), this encoder explains a large proportion of NSD voxel variance (typically $R^2 > 0.6$ in high-level ROIs), validating its use as a reliable neural readout. We also aaply this predictor to EEG data and it can still make reliable prediction with $R^2 \approx 0.45$. Because the encoder is not CLIP-based, it provides an independent estimate of neural activity.

## 4.3 RESULTS

### 4.3.1 GENERATE PREFERRED REPRESENTATIONS IN HUMAN BRAIN

With structured landscape, we can get a better representation of certain brain region by sampling around the prior in latent space, and generate images with those latent values, as shown in Figure 1.

In our latent space, we interpolate each latent code $z$ toward a positive or negative prior $z' = \alpha \cdot \mu_{\text{prior}} + (1 - \alpha) \cdot z$—and pass it to a frozen SDEdit diffusion model with $\gamma = 1.0$; thus the output image depends solely on the edited embedding, allowing a clean comparison with the baseline.

Figure 3 and Table 1 showed results on NSD dataset, our method could changes activations towards correct direction in all regions, while baseline method makes mistakes. Also, our method produces greater decrease in activation in all regions, demonstrating effectiveness of defining an independent negative prior when constructing landscape rather than simply reverse the maximal direction in baseline. For increasing activation, our method outperforms the baseline in not well-studied regions like SPL, IPS, TE, IP, 31, and might be attributed to their mix-selectivity (Vialatte et al., 2020; Taylor & Xu, 2024; Maranesi et al., 2024). In highly highly category-selective regions like FFA and OPA (Downing et al., 2001; Kanwisher et al., 1999; McCarthy et al., 1997). Our method achieves comparable performance with baseline. Our method also consistently produces more realistic visual representations in all regions with lower FID scores (Heusel et al., 2017), that sampling from a structured latent space in VAE may be more controllable than perturbing CLIP embeddings directly.

Table 1: Comparison of activation change and realism between our method and the baseline. Better values are in **bold face**. For activation changes, ↑ indicates that higher values are better, and ↓ indicates that lower values are better. (a) Results on the NSD dataset. (b) Results on the THINGS-EEG dataset. Our method has a consistent direction for activation change as targeted direction.

**(a) NSD Results**

| | ROIs | TE2p | IP0 | IP1 | OFA | FFA | EBA | OPA |
|---|---|---|---|---|---|---|---|---|
| Activation Increase (↑) | Our Method | **0.242** | **0.498** | **0.153** | 1.052 | 0.891 | 0.963 | **0.699** |
| | BrainACITV | 0.207 | 0.463 | 0.044 | **1.155** | **0.976** | **1.160** | **0.699** |
| Activation Decrease (↓) | Our Method | **-0.136** | **-0.699** | **-0.073** | **-0.660** | **-0.580** | **-0.810** | **-0.824** |
| | BrainACTIV | 0.149 | 0.186 | 0.092 | 0.213 | -0.484 | -0.669 | -0.172 |
| FID @ 2k (↓) | Our Method | **172.71** | **173.34** | **178.01** | **173.66** | **153.70** | **137.33** | **166.01** |
| | BrainACTIV | 225.77 | 213.17 | 200.14 | 216.15 | 256.44 | 188.59 | 225.37 |

**(b) THINGS-EEG Results**

| | Channels | O1 | Oz | O2 | PO7 | PO3 | POz | PO4 | PO8 |
|---|---|---|---|---|---|---|---|---|---|
| Activation Increase (↑) | Our Method | **0.079** | **0.146** | **0.085** | **0.171** | **0.116** | **0.111** | 0.060 | **0.072** |
| | BrainACTIV | -0.003 | 0.138 | 0.005 | 0.153 | -0.019 | 0.085 | **0.083** | 0.058 |
| Activation Decrease (↓) | Our Method | **-0.066** | -0.134 | -0.132 | **-0.137** | **-0.085** | -0.212 | -0.031 | **-0.079** |
| | BrainACTIV | 0.001 | **-0.298** | **-0.202** | -0.123 | -0.072 | **-0.261** | **-0.050** | -0.027 |
| FID @ 2k (↓) | Our Method | **120.20** | **99.54** | **108.59** | **109.07** | **123.41** | **107.10** | **106.23** | **117.18** |
| | BrainACTIV | 157.25 | 159.90 | 171.77 | 164.34 | 133.81 | 140.48 | 164.24 | 178.66 |

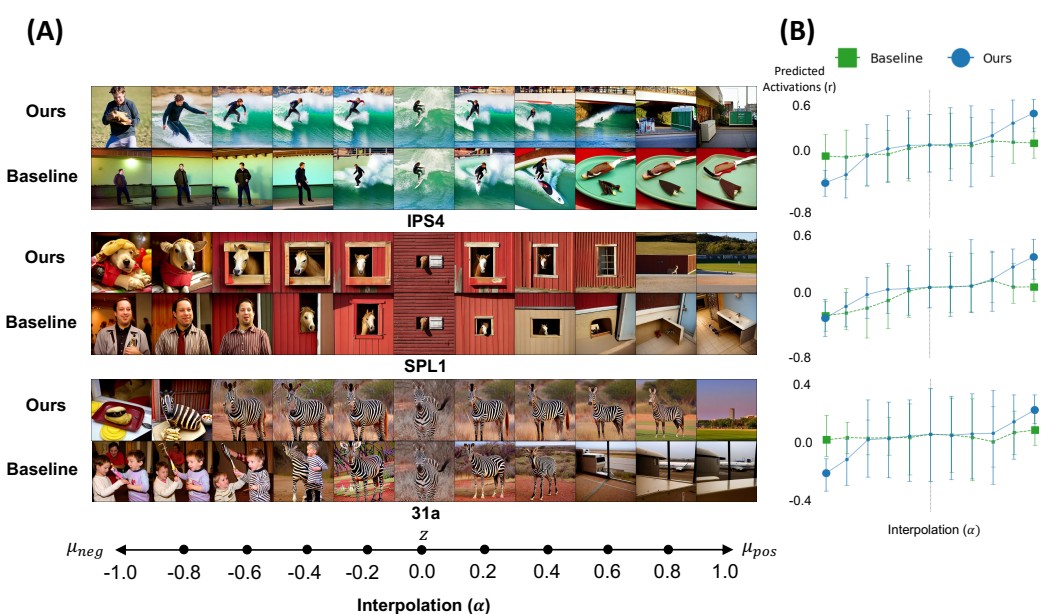

Figure 3: **Activation change results of our method and baseline.** **(A)** Generate preferred representations of multiple ROIs with example images from COCO dataset by our method and baselines. The alpha value in interpolation indicates the strength and direction of preferred semantic in generation. **(B)** Comparison between predicted activations of ROIs made by DINO-ViT encoder. Means and ranges of activations are presented for different alpha settings. Our method outperforms the baseline in terms of maximizing or minimizing the ROI activations.

For THINGS-EEG results in Table 1, our model also consistently produces results with correct signs across the occipital channels in dataset, outperforms baseline. Our method generates representations that better activate the majority of channels and always produces more realistic representations. The results show that our method can be generalized to multi-modal neuronal data.

Besides, to demonstrate the robustness of our predictor, we include a comparison between activation modulation results of original subjects and a held-out subject from NSD dataset. The results are shown in Table 2. The comparable predictions prove the reliability of our results.

Table 2: Cross-subject generalization result on NSD dataset. The table compares the predicted activation change on original subject data versus a held-out subject across several visual ROIs.

| ROIs | OFA | FFA | EBA | VWFA | OPA | PPA | RSC |
|---|---|---|---|---|---|---|---|
| Activation increase (original subj) | 0.869 | 0.851 | 0.860 | 0.867 | 0.731 | 0.966 | 0.942 |
| Activation increase (held-out subj) | 0.861 | 0.975 | 0.790 | 0.648 | 0.629 | 1.061 | 0.940 |
| Activation decrease (original subj) | -0.526 | -0.382 | -0.564 | -0.279 | -0.557 | -0.610 | -0.775 |
| Activation decrease (held-out subj) | -0.436 | -0.302 | -0.391 | -0.089 | -0.485 | -0.813 | -0.823 |

### 4.3.2 GENERATE PREFERRED REPRESENTATIONS IN ARTIFICIAL NEURAL NETWORKS

Besides biological brains , we also test our approach on purely *artificial* targets: a mid-level unit inside a SimCLR encoder (Chen et al., 2020) a contrastive-learning framework with a ResNet-50 backbone, as well as a unit from a more complex Vision Transformer (ViT). Fig. 4 shows natural images that have the largest responses from the chosen units in ANNs. With our model trained on unit activations, editing latent to the positive prior could generate representations that can better activate the given ANN units compared with the baseline as shown in the boxplot.

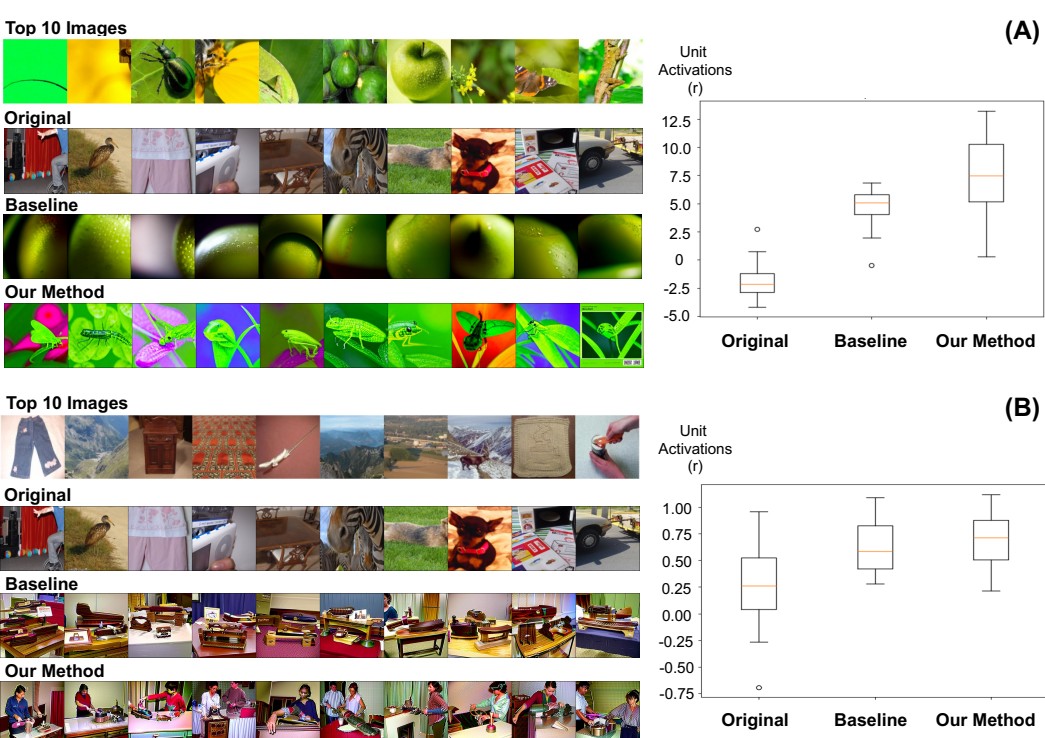

Figure 4: **Generate preferred images with increasing unit activations in ANNs.** **(A)** The top 10 images demonstrate a notable pattern of "green plant & insect" represented by a unit in the middle layer of SimCLR model. our method generates images that strongly match the pattern and increase activation more than the baseline. **(B)** For a mid-layer unit in a more complex Vision Transformer (ViT), our method synthesizes images that can better activate the unit compared with baseline, though the concept seems to be ambiguous.

### 4.3.3 OPTIMIZE UNIT ACTIVATIONS WHILE PRESERVING UNRELATED FEATURES

To demonstrate the strength of our structured VAE latent space, we trained two lightweight image-to-image VAEs on the CIFAR-10 dataset. a standard VAE with a Gaussian prior, and our proposed VAE whose latent space is regularized by activations from a pre-trained ConvNet. We constrain VAE latent with multiple units activations, and only edit the latent dimensions controlled by a certain unit. Beside direct modification towards preset priors, we optimize latent with Adam

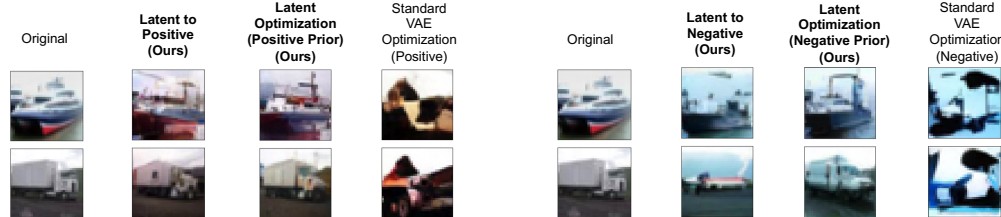

Figure 5: **Generating preferred images with different methods on CIFAR-10. Left 4 columns** show optimizing images to increase their activations. **Column 1** are two example input images. **Column 2-3** are results of our method while **column 4** represent optimizing the latent space with standard VAE. **Right 4 columns** are results of optimization with same methods with target of decreasing activations.

Table 3: Summary of average cosine similarity and unit responses change ($\Delta r$) between new images and input images over different optimization methods, corresponding to Figure 5. The best results are marked with **bold face**.

| Direction | Model | Optimization Method | Average Similarity (Cosine) | Average $\Delta r$ |
|---|---|---|---|---|
| positive | structured landscape | latent modification | **0.745** | 0.046 |
| | | gradient descent | 0.666 | **0.079** |
| | original landscape | gradient descent | 0.520 | 0.067 |
| negative | structured landscape | latent modification | **0.696** | -0.083 |
| | | gradient descent | 0.661 | **-0.110** |
| | original landscape | gradient descent | 0.538 | -0.068 |

optimizer to maximize or minimize the activation of targeted units by combining the activation with a KLD regularization term as target, ensuring the latent code remains aligned with the prior distribution. The results of latent modification and optimization on both VAEs are shown in Figure 5. In Table 3, we measure similarity between original and new images with a pre-trained ResNet18 (He et al., 2016). Those results indicate that our structured landscape approach not only achieves more robust activation changes but also maintains a higher similarity to the original images compared to standard VAE, underscoring the efficacy of incorporating structured priors for latent optimization.

## 5 Discussion and Limitations

We developed Vis-Lens, a framework that imposes a novel activation-regularized priors on VAE to yield structured landscape and then performs direct latent editing to steer activations in biological and artificial networks. This structured landscape can also modulates the activation of target units while preserving unrelated features. Vis-Lens generates visual representations that show stronger activation modulation ability and better realism on brain regions compared with state-of-the-art method, and also proved to be more effective on feature visualization in artificial neural networks.

There are also limitations for our work. Firstly, the image generator and the CLIP encoder are frozen, any biases inherited from their pre-training data might propagate into our newly generated preferred images. Also, the reliability of our results is limited due to the complete in-silico framework design.

In the future, our work may focus on scaling the framework to more data modalities like MEG or larger scale brain datasets. Meanwhile, we aim to proceed beyond evaluations with in-silico human brain simulators, and test the generated preferred images in human experiments; such closed-loop validation would further confirm the effectiveness of our framework and open the door to a broader range of brain computer interface applications.

## ETHICS STATEMENT

This research was conducted on datasets which are all publicly available and no ethics concerns, no new data was collected from human subjects. Besides, all experiments in our study are purely in-silico. The goal of our work is to advance the scientific understanding of neural information, and we do not foresee any direct societal risks or negative ethical concerns arising from our methods.

## REPRODUCIBILITY STATEMENT

All details for data preprocessing, model training, activation analysis, and figure generation is available in the main text and supplementary materials and will be released on GitHub upon publication. All data used in this research can be accessed from public sources. The computational environment and all model hyperparameters required to reproduce our main experimental results are detailed in the appendix.

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

APPENDIX

THE USE OF LARGE LANGUAGE MODELS

We did not use Large Language Models for this paper.

APPENDIX OVERVIEW

This appendix provides comprehensive details and additional results to support the findings presented in the main paper and ensure that all of the results can be reproduced. The appendix content is organized into the following sections:

- **Additional Results**: Additional visualizations and analyses that further demonstrate the effectiveness of our proposed methods. More examples of enhanced sampling, local change results, and optimization outcomes are presented, as shown in section A.
- **Experimental Settings**: Comprehensive information about the datasets settings, like input size and how are the datasets divided and processed, model architectures and implementation details, then training configurations and computational resources. All of them can provide a clear understanding of our experimental setup, which are helpful to reproduce the results, as shown in section B.
- **Implementation Details**: Detailed information about the algorithms we used to implement our method, including pseudocode, and corresponding explanations. All of them can provide a clear understanding of how the training and optimizaztion are carried out, as shown in section C.
- **Predictive Power of In-silico brain simulator**: Information about how much our trained In-silico brain simulator can predict the brain activity data, as shown in section D.

## A  ADDITIONAL RESULTS

### A.1  MORE RESULTS FOR THE NSD DATASET

Here is a summarized table of more experimental results on NSD dataset:

Table S1: Activation change and realism on additional ROIs. Better values are in **bold**.

| | ROIs | V1 | V2 | V3 | V4 | V8 | LO1 | LO2 | V4t |
|---|---|---|---|---|---|---|---|---|---|
| **Activation Increase** ($\Delta r$) | Ours | **0.257** | 0.080 | **0.247** | 0.486 | 0.696 | **0.583** | 0.498 | 0.757 |
| | Baseline | 0.157 | **0.116** | 0.226 | **0.488** | **0.721** | 0.582 | **0.517** | **0.767** |
| **Activation Decrease** ($\Delta r$) | Ours | **-0.291** | **-0.272** | **-0.573** | **-0.690** | **-0.406** | **-0.485** | **-0.490** | **-0.601** |
| | Baseline | 0.102 | 0.224 | 0.274 | 0.501 | 0.608 | 0.100 | -0.057 | -0.511 |
| **FID@2k** | Ours | **164.87** | **157.84** | **167.44** | **173.24** | **160.07** | **136.05** | **186.67** | **178.62** |
| | Baseline | 224.29 | 201.56 | 240.64 | 231.60 | 216.80 | 208.23 | 255.06 | 197.61 |

| | ROIs | VWFA-1 | PPA | RSC | OWFA | IP2 | TPOJ1 | TPOJ2 | TPOJ3 |
|---|---|---|---|---|---|---|---|---|---|
| **Activation Increase** ($\Delta r$) | Ours | **1.108** | **0.952** | 0.923 | 0.718 | **0.091** | 0.456 | 0.661 | 1.014 |
| | Baseline | 1.019 | 0.926 | **0.954** | **1.088** | 0.084 | **0.509** | **0.751** | **1.091** |
| **Activation Decrease** ($\Delta r$) | Ours | **-0.391** | **-0.809** | **-0.913** | **-0.780** | **-0.213** | **-0.136** | **-0.384** | **-0.591** |
| | Baseline | 0.350 | -0.579 | -0.804 | 0.347 | 0.061 | -0.134 | -0.234 | -0.484 |
| **FID@2k** | Ours | **169.14** | **153.67** | **172.92** | **163.88** | **168.56** | **192.04** | **131.83** | **137.57** |
| | Baseline | 229.65 | 234.79 | 216.65 | 207.15 | 171.94 | 244.44 | 197.94 | 197.24 |

The full test-set results for the additional ROIs are provided in Figs. S1–S16.

### A.1.1  ROI: IPS4

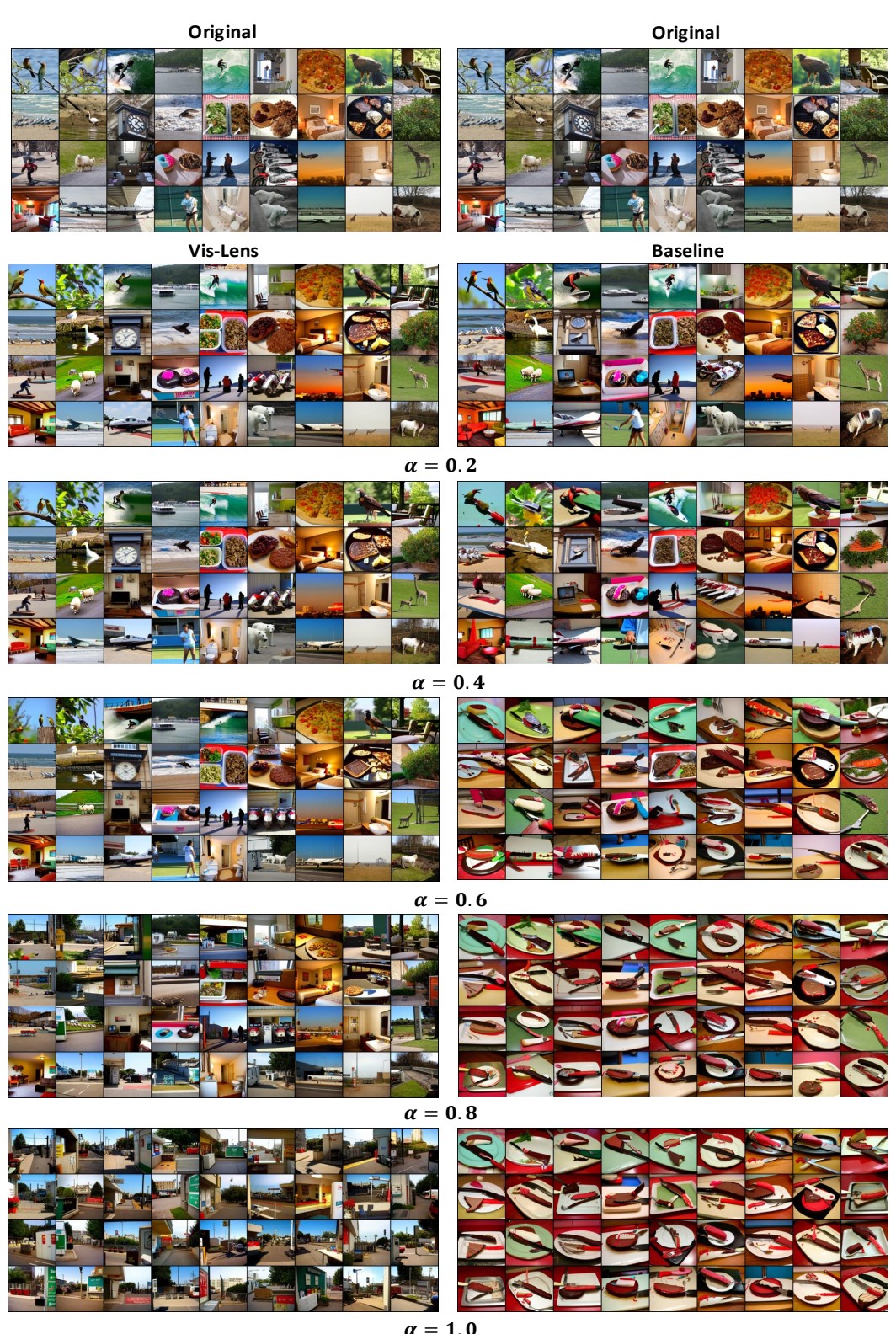

Figure S1: Full result of activation change on region IPS4, postive direction.

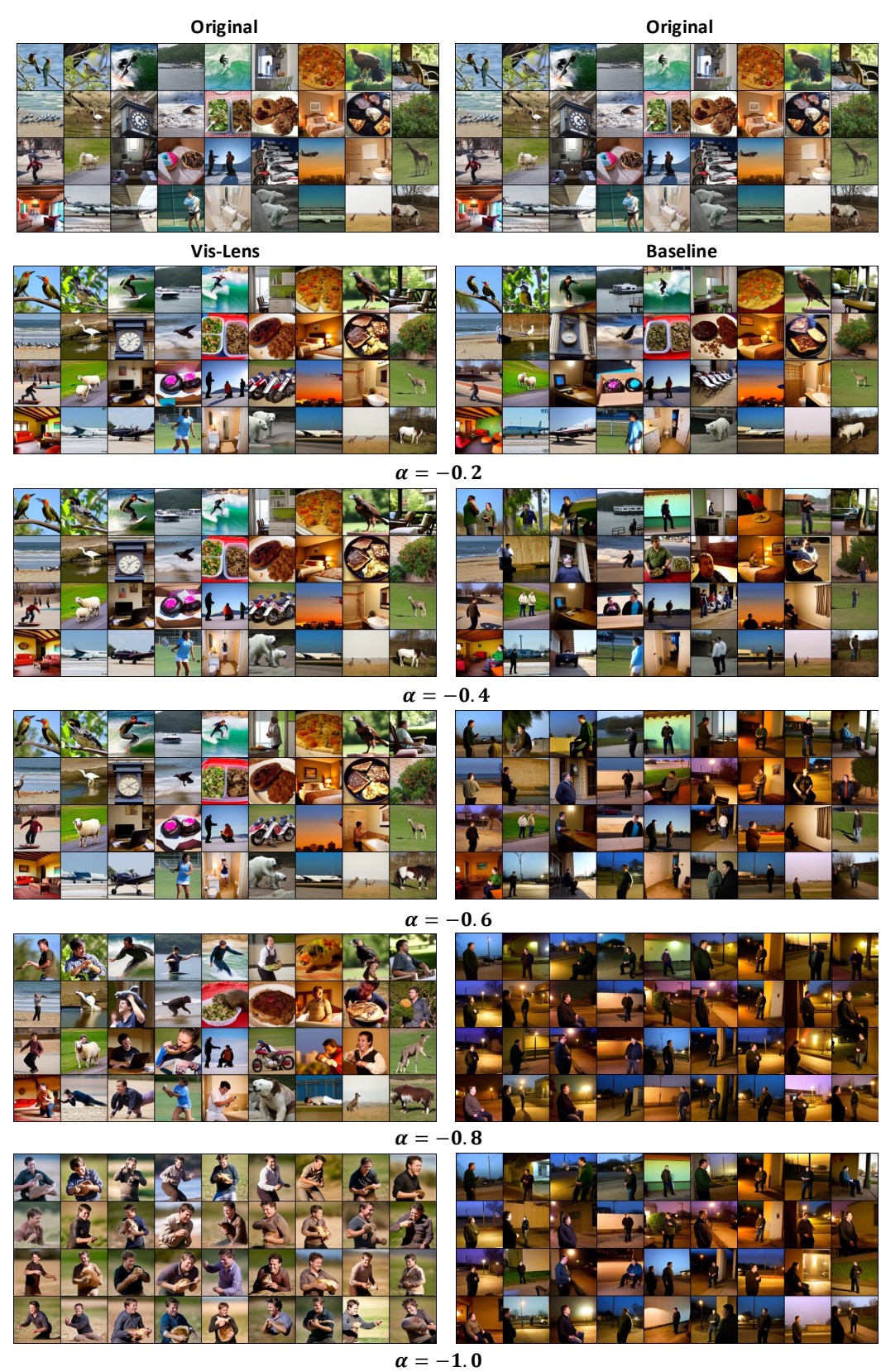

Figure S2: Full result of activation change on region IPS4, negative direction.

A.1.2    ROI: SPL1

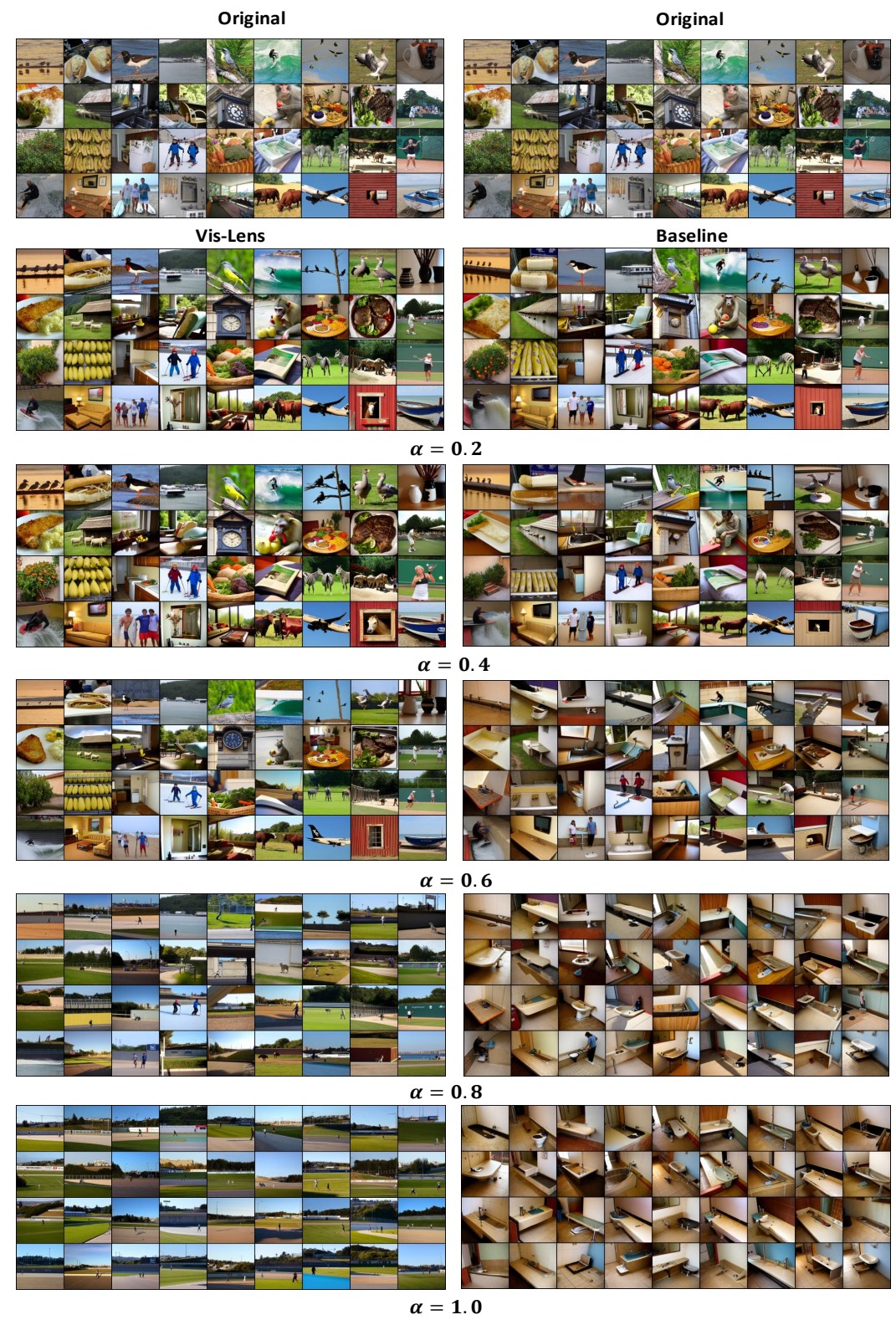

Figure S3: Full result of activation change on region SPL1, positive direction.

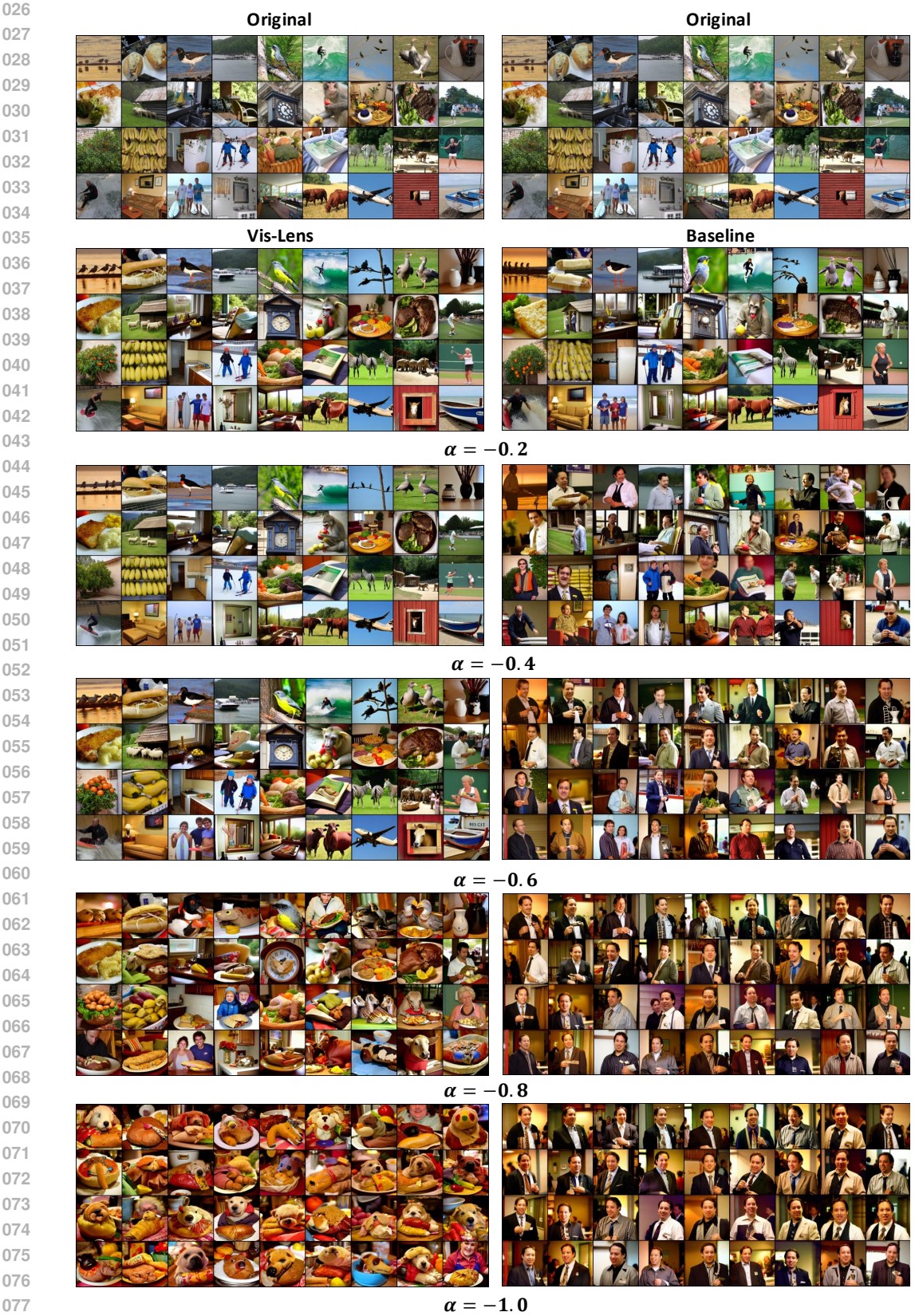

Figure S4: Full result of activation change on region SPL1, negative direction.

### A.1.3   ROI: 31A

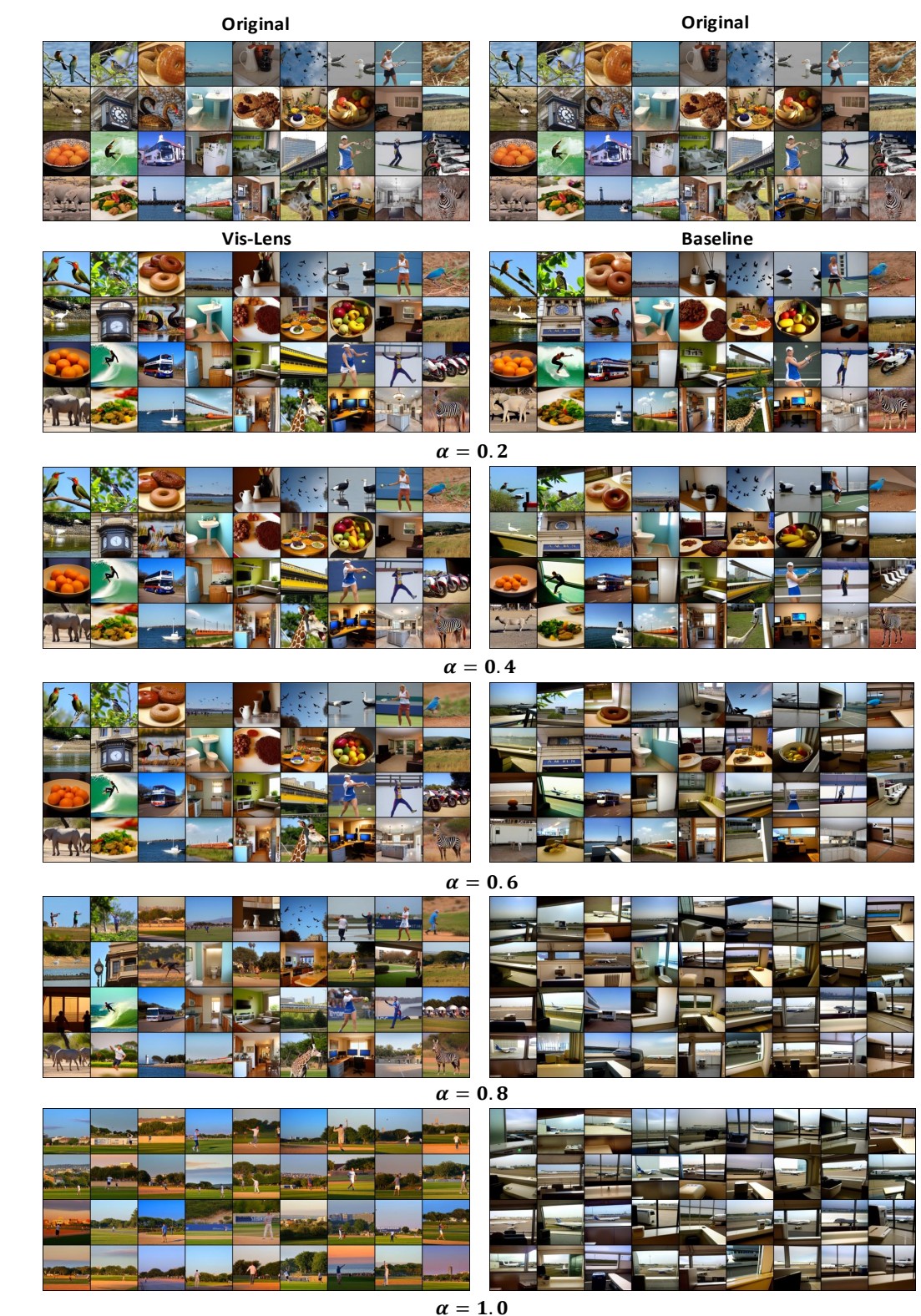

Figure S5: Full result of activation change on region 31a, positive direction.

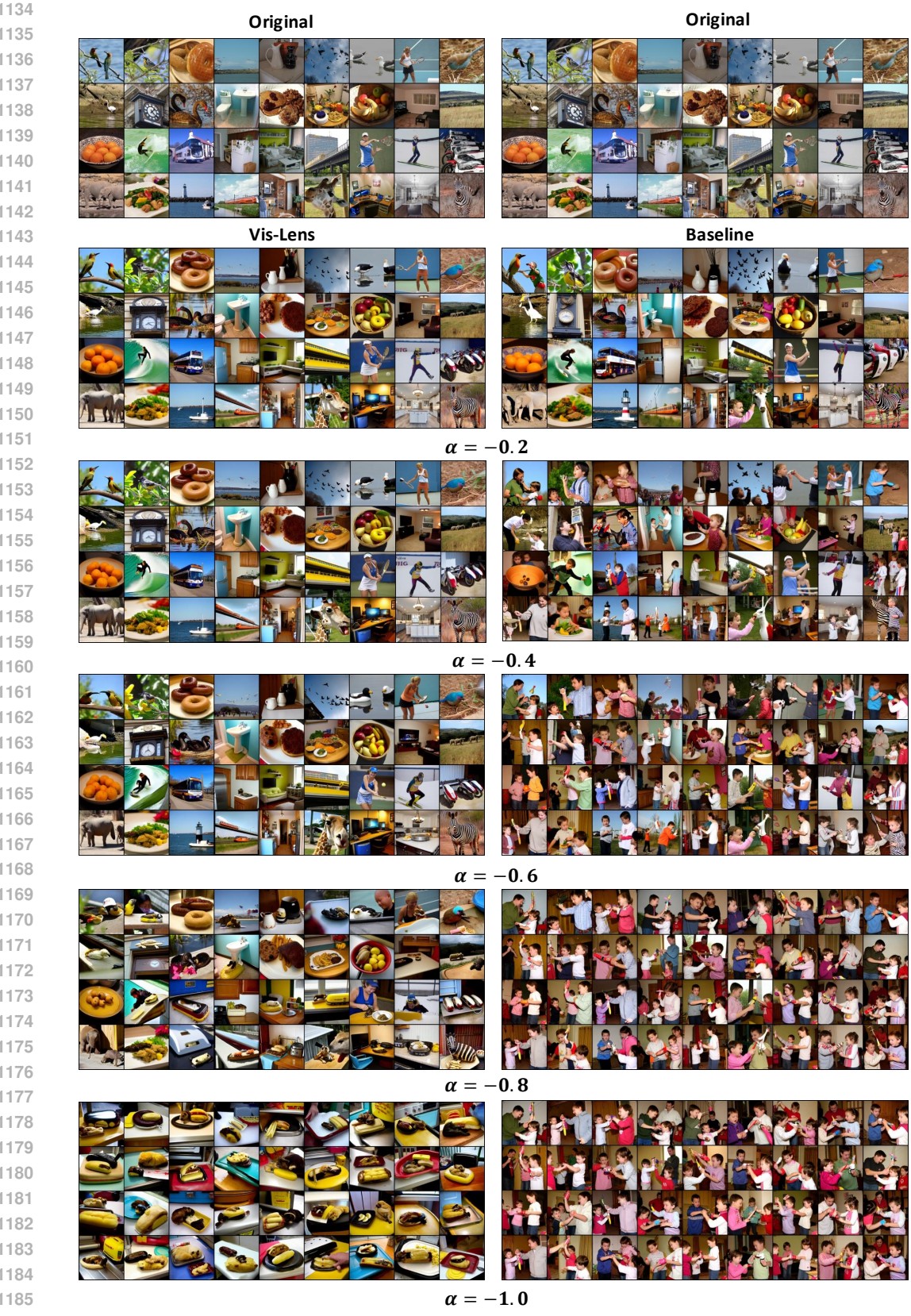

Figure S6: Full result of activation change on region 31a, negative direction.

## A.1.4 ROI: 31PD

**Original**

**Original**

**Vis-Lens**

**Baseline**

$\alpha = 0.2$

$\alpha = 0.4$

$\alpha = 0.6$

$\alpha = 0.8$

$\alpha = 1.0$

Figure S7: Full result of activation change on region 31pd, positive direction.

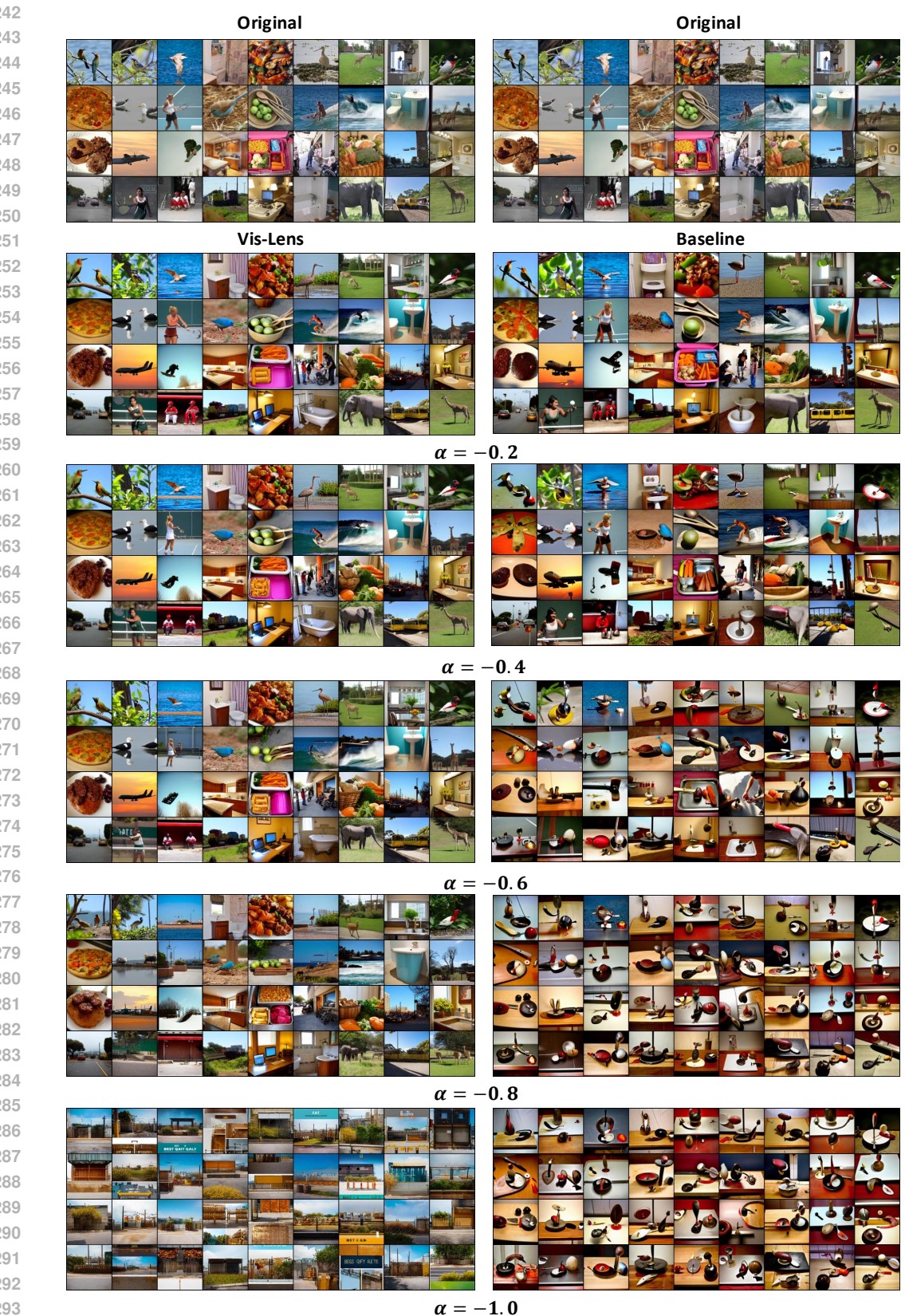

Figure S8: Full result of activation change on region 31pd, negative direction.

## A.1.5 ROI: V1

Figure S9: Full result of activation change on region V1, positive direction.

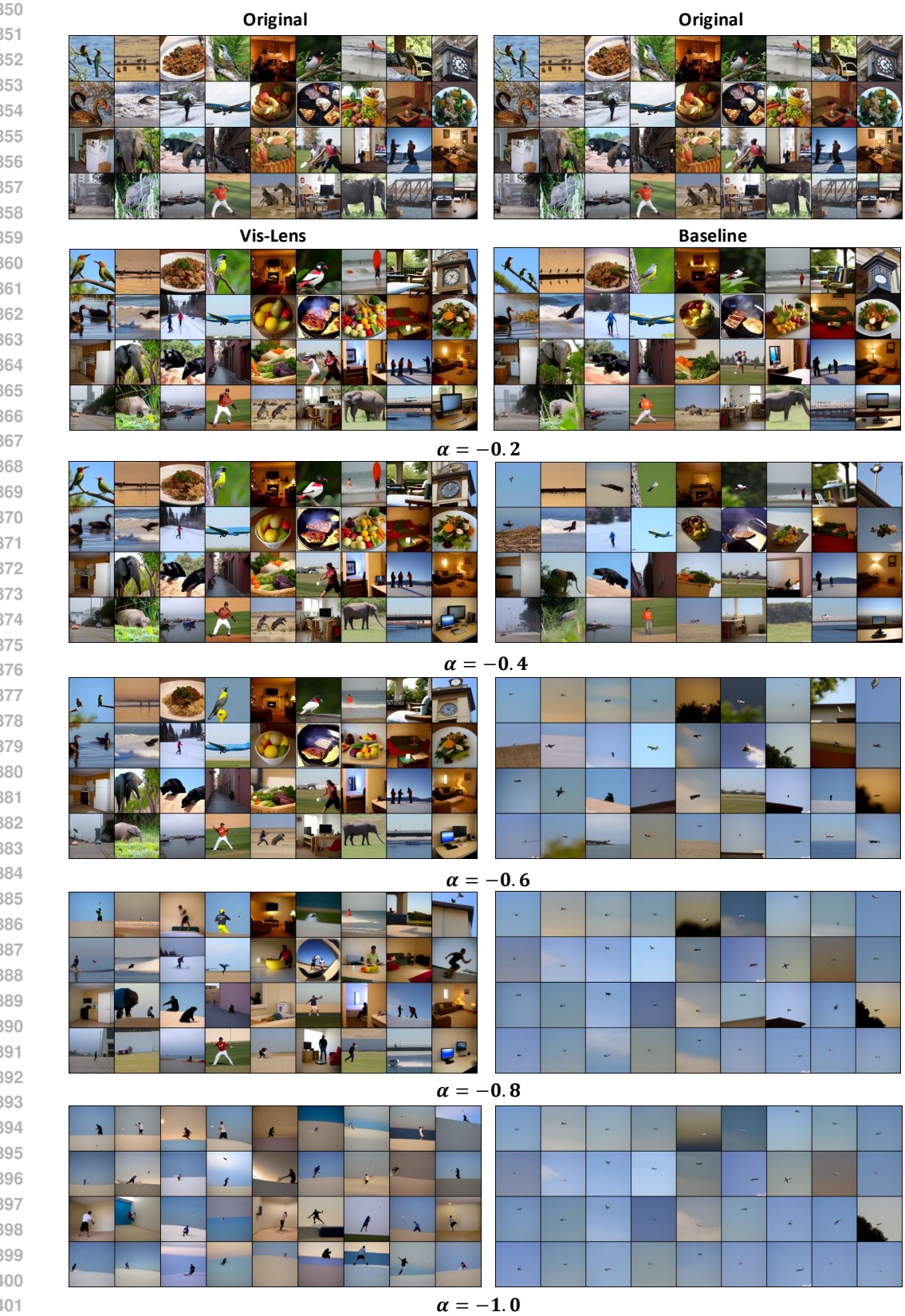

Figure S10: Full result of activation change on region V1, negative direction.

A.1.6 ROI: V2

**Original**

**Original**

**Vis-Lens**

**Baseline**

$\alpha = 0.2$

$\alpha = 0.4$

$\alpha = 0.6$

$\alpha = 0.8$

$\alpha = 1.0$

Figure S11: Full result of activation change on region V2, positive direction.

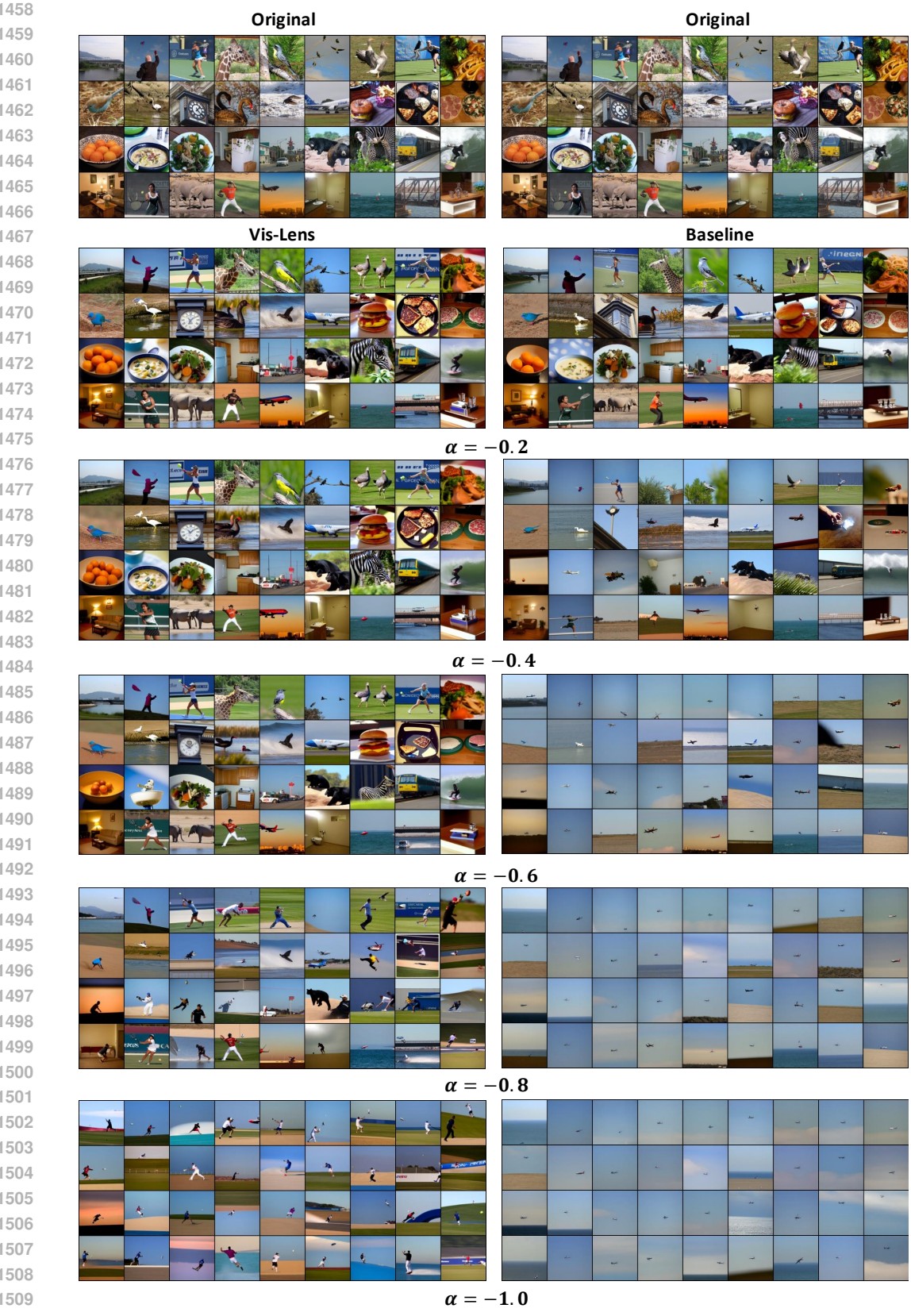

Figure S12: Full result of activation change on region V2, negative direction.

### A.1.7 ROI: V3

**Original**

**Original**

**Vis-Lens**

**Baseline**

$\alpha = 0.2$

$\alpha = 0.4$

$\alpha = 0.6$

$\alpha = 0.8$

$\alpha = 1.0$

Figure S13: Full result of activation change on region V3, positive direction.

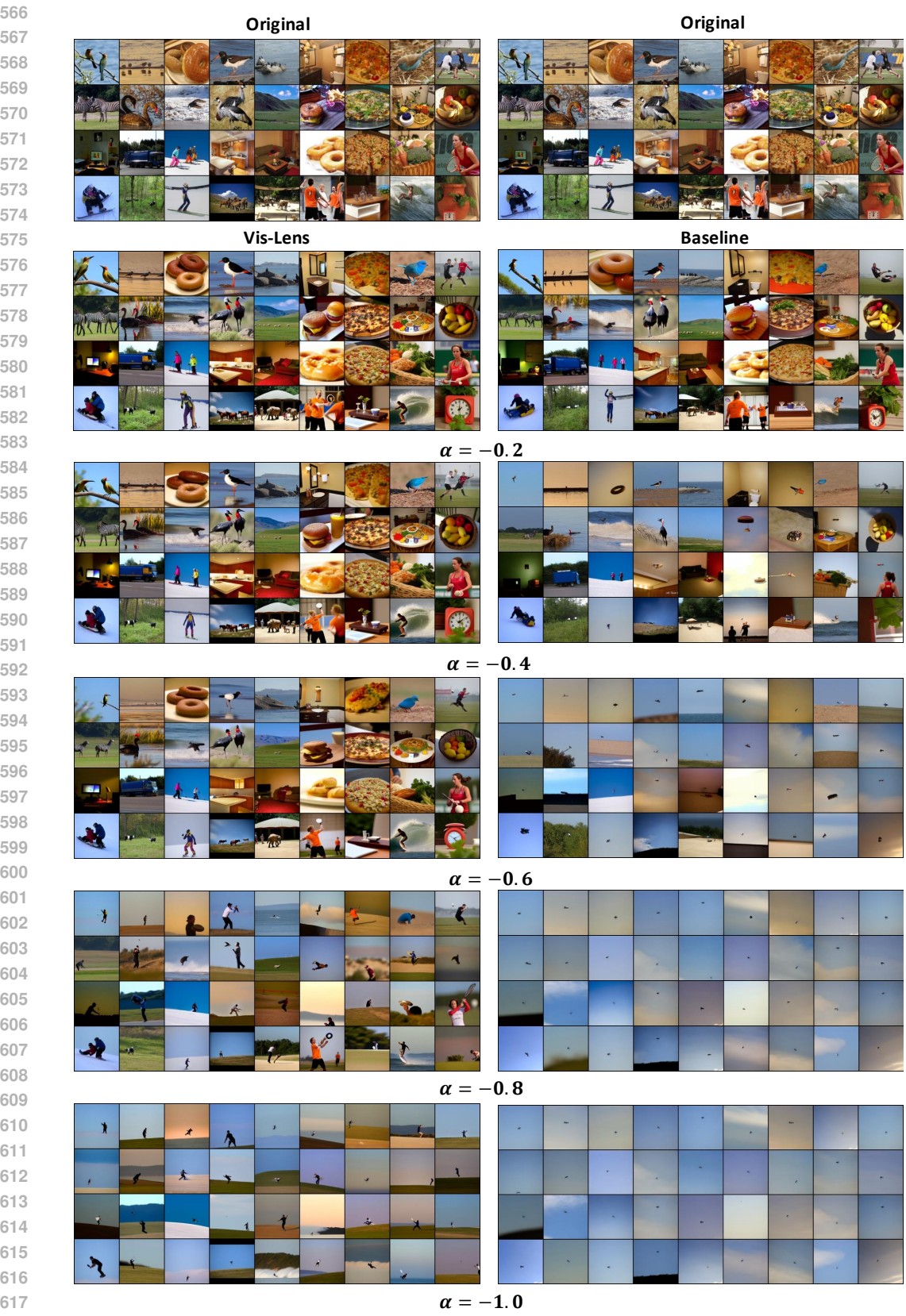

Figure S14: Full result of activation change on region V3, negative direction.

A.1.8    ROI: V4

**Original**

**Original**

**Vis-Lens**

**Baseline**

$\alpha = 0.2$

$\alpha = 0.4$

$\alpha = 0.6$

$\alpha = 0.8$

$\alpha = 1.0$

Figure S15: Full result of activation change on region V4, positive direction.

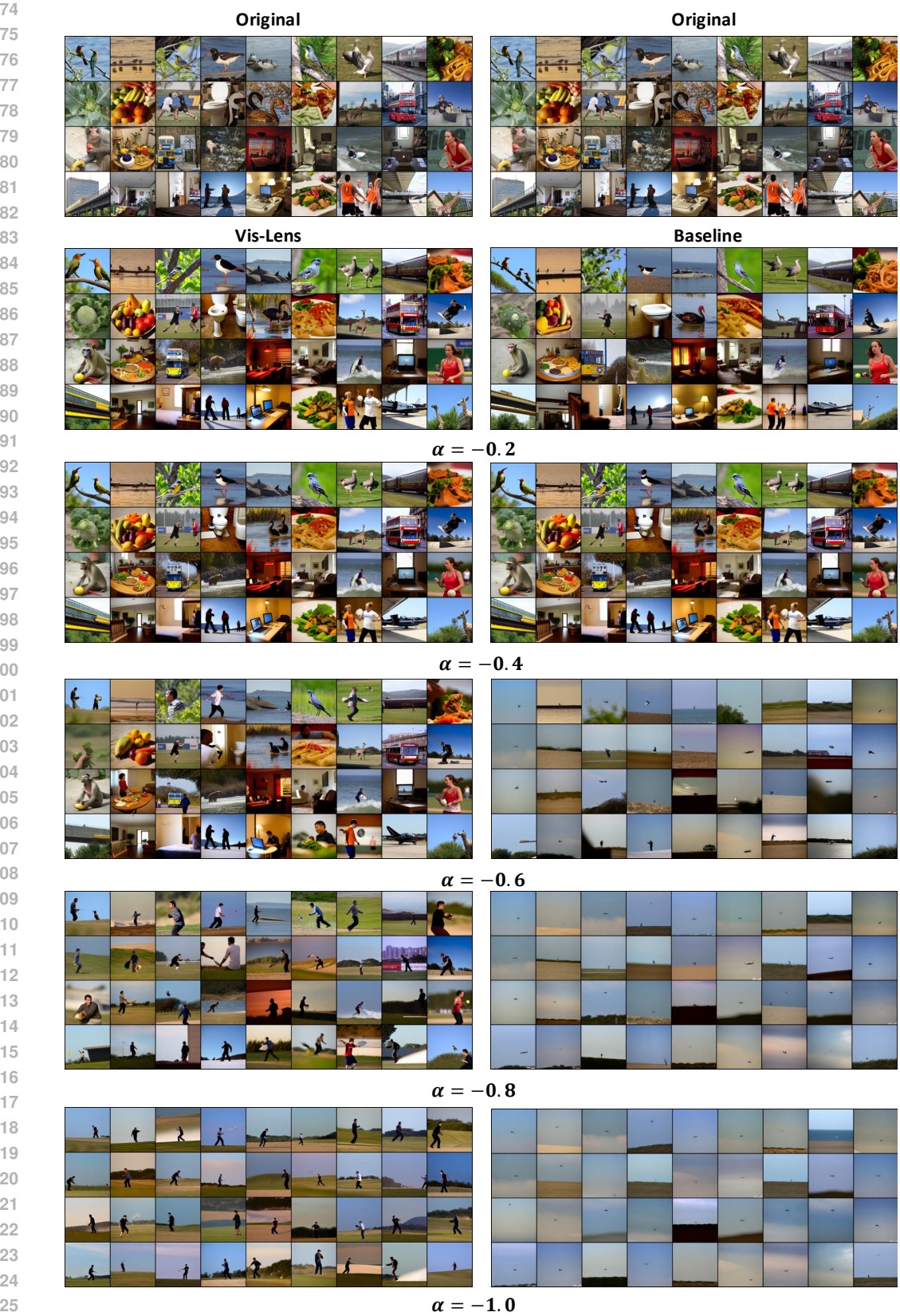

Figure S16: Full result of activation change on region V4, negative direction.

## A.2 VISUAL REPRESENTATIONS OF EEG OCCIPITAL CHANNELS

Here we present the visual representations generated by our method. 'Original' means input images, 'Reconstruction' means image reconstructed with our model with given input, 'Maximize' and 'Minimize' represent images synthesized by out method to increase/decrease activations of a given channel.

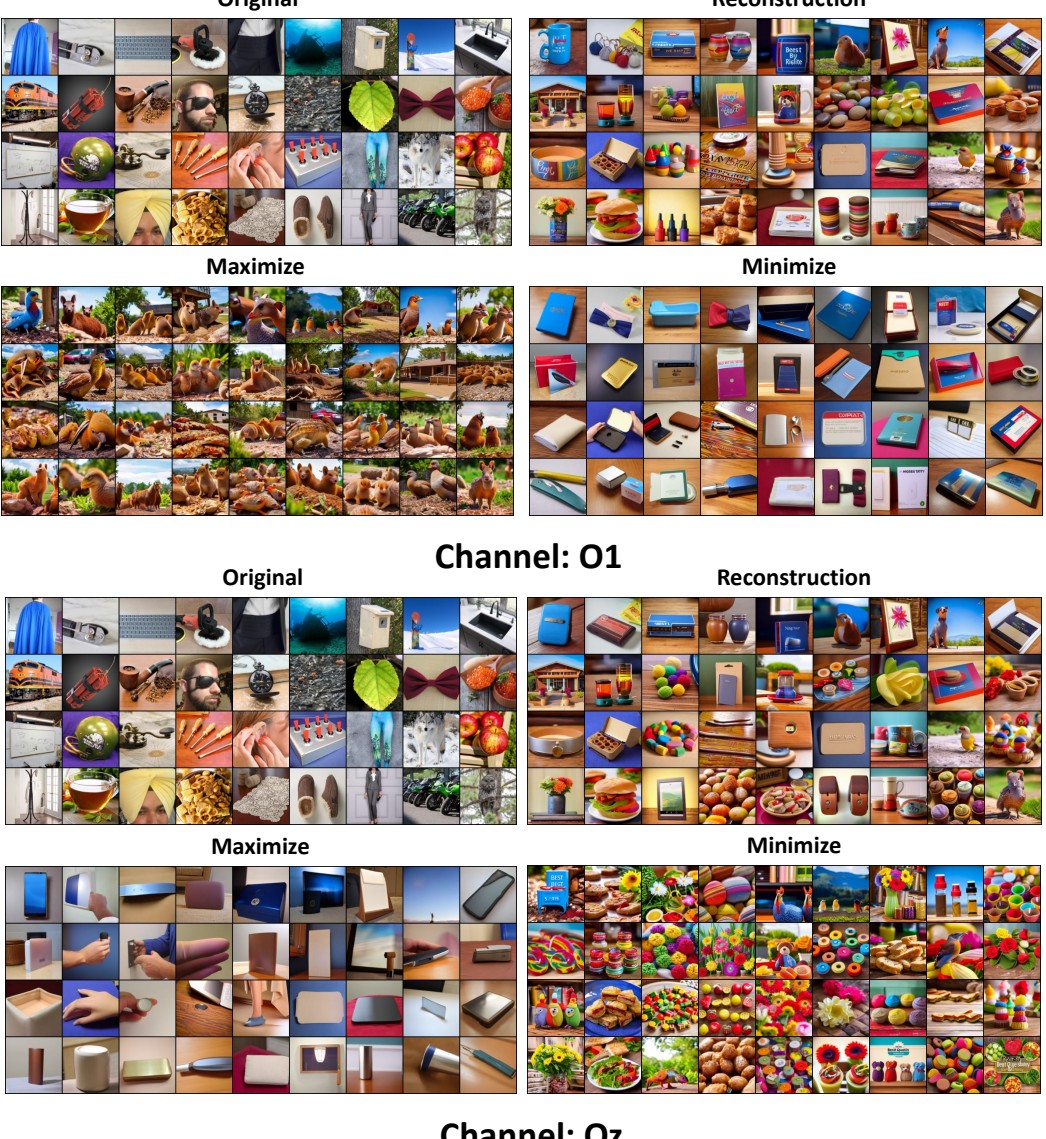

Figure S17: Result of activation change on channel O1 & Oz with our method

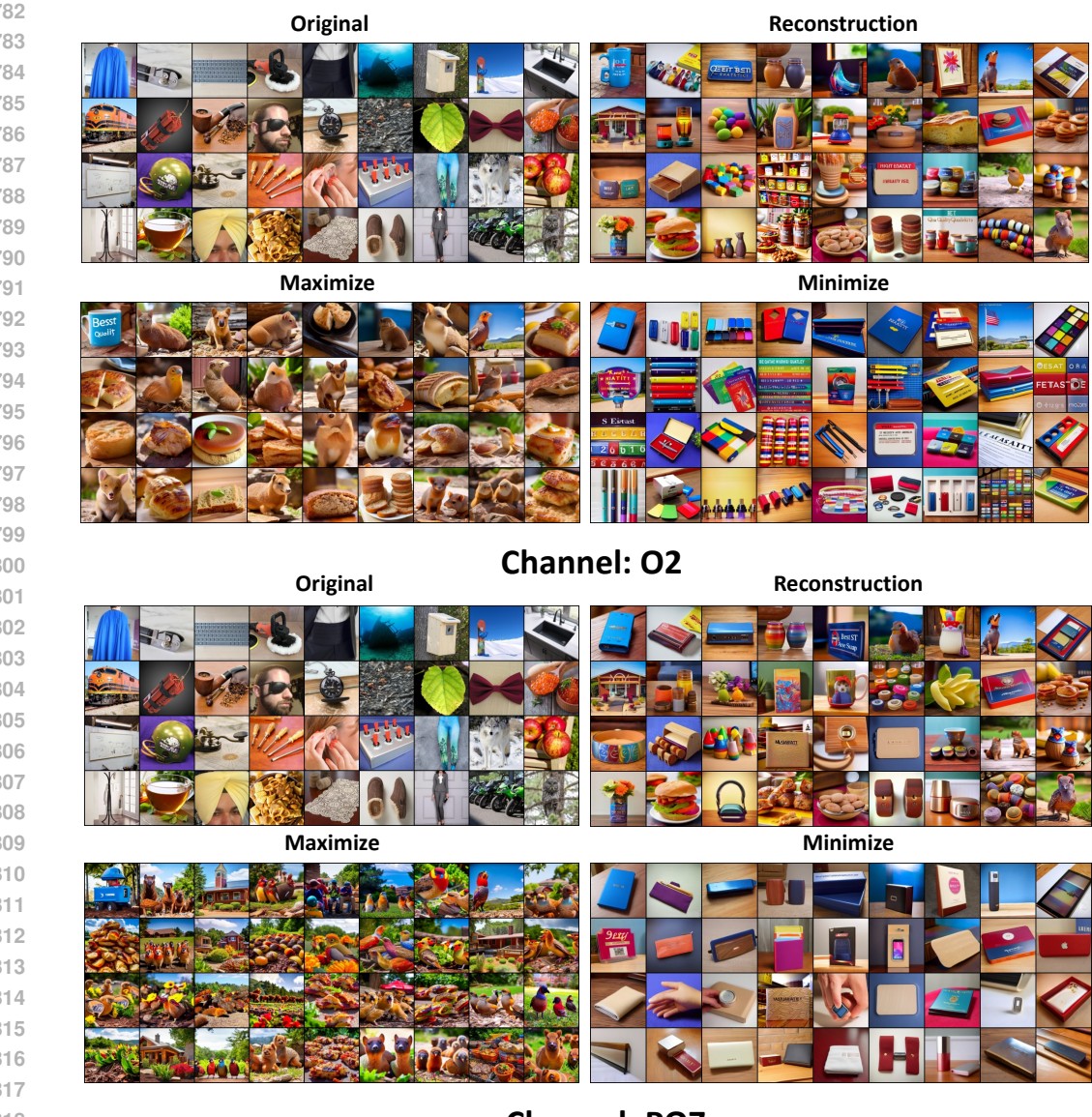

Figure S18: Result of activation change on channel O2 & PO7 with our method

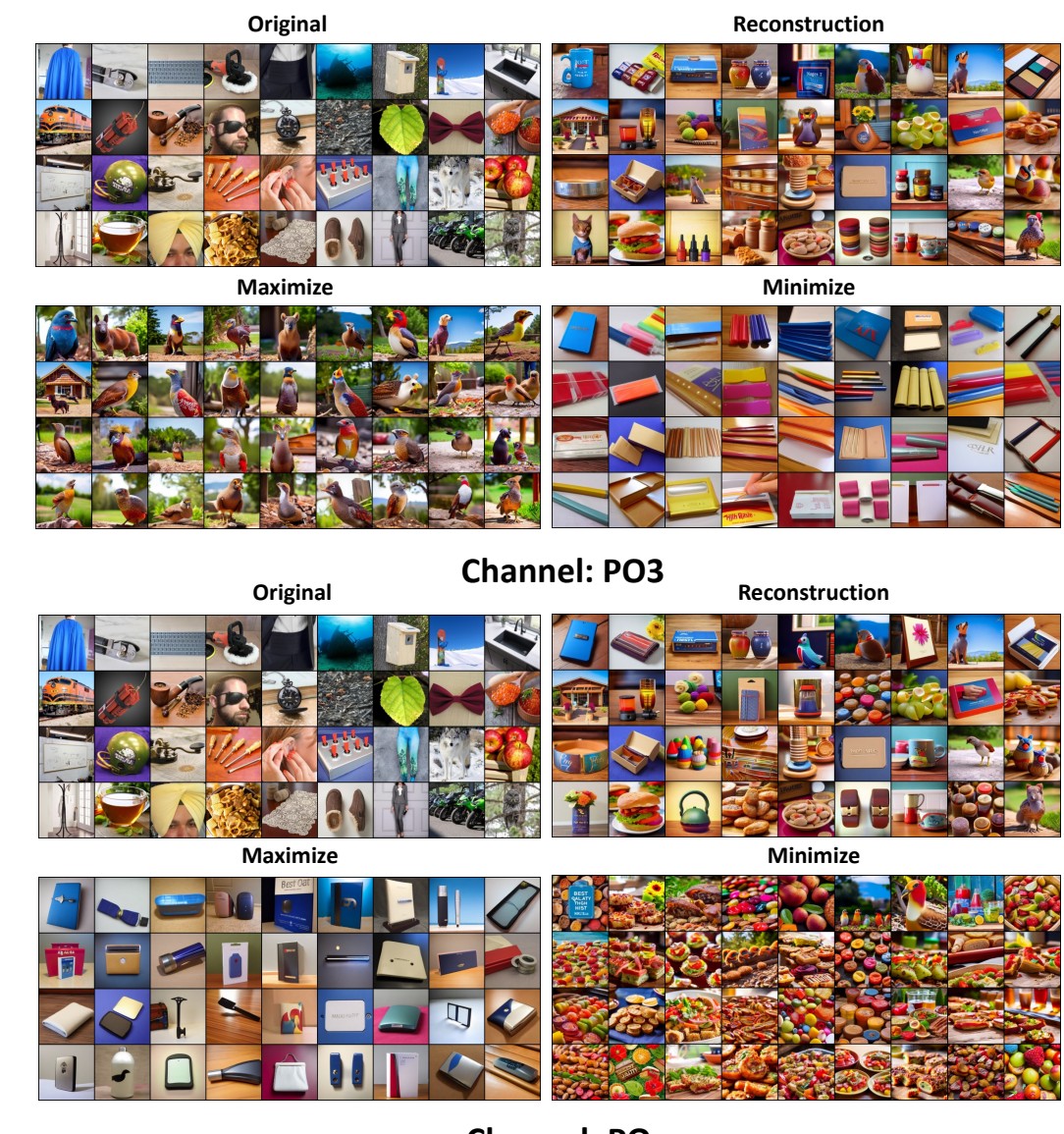

Figure S19: Result of activation change on channel PO3 & POz with our method

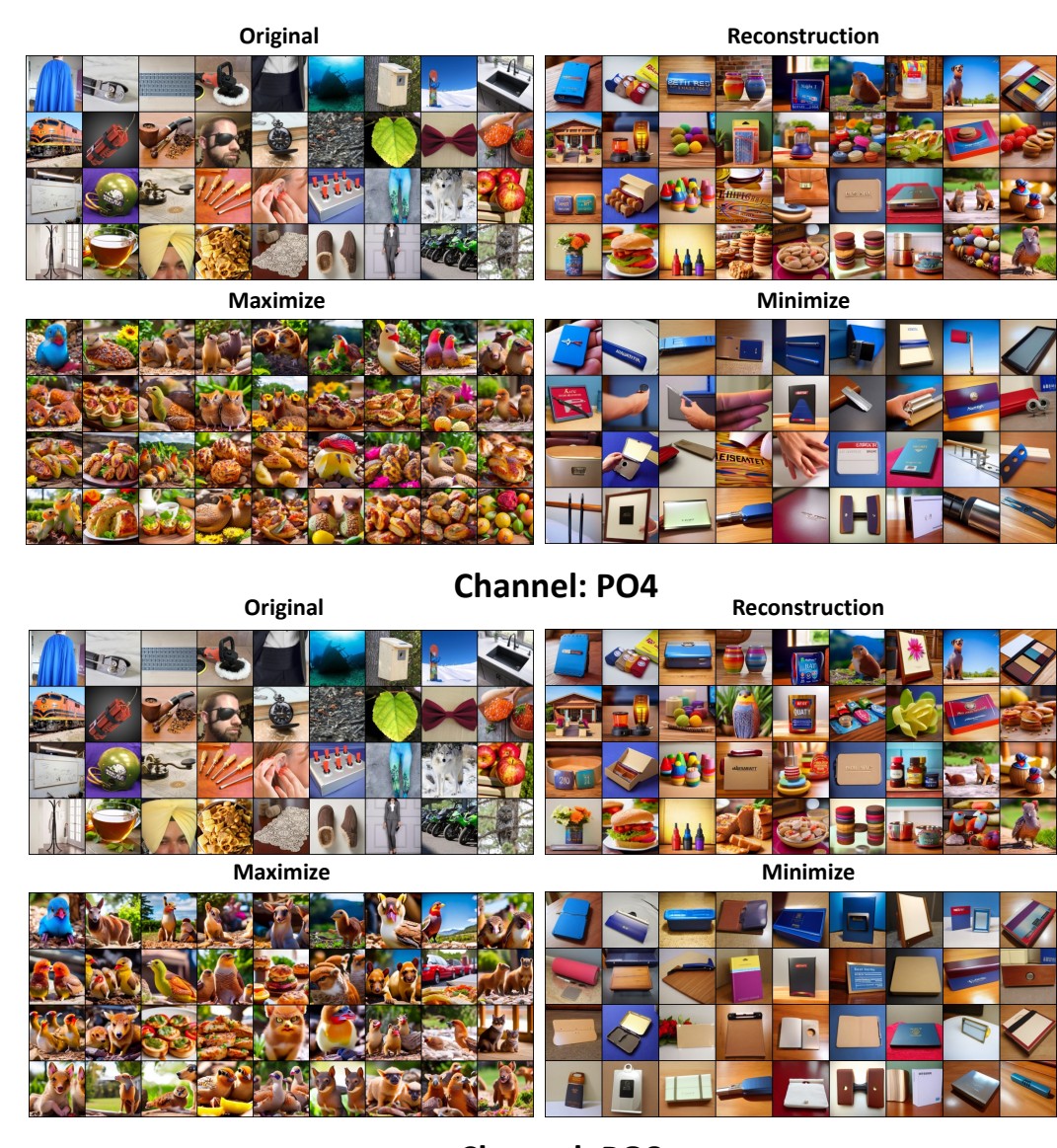

Figure S20: Result of activation change on channel PO4 & PO8 with our method

### A.3 MORE ANN RESULTS

We present more examples on manipulating activations in SimCLR(Chen et al., 2020), shown in Figs. S17–S22, which further demonstrate the effectiveness of our method.

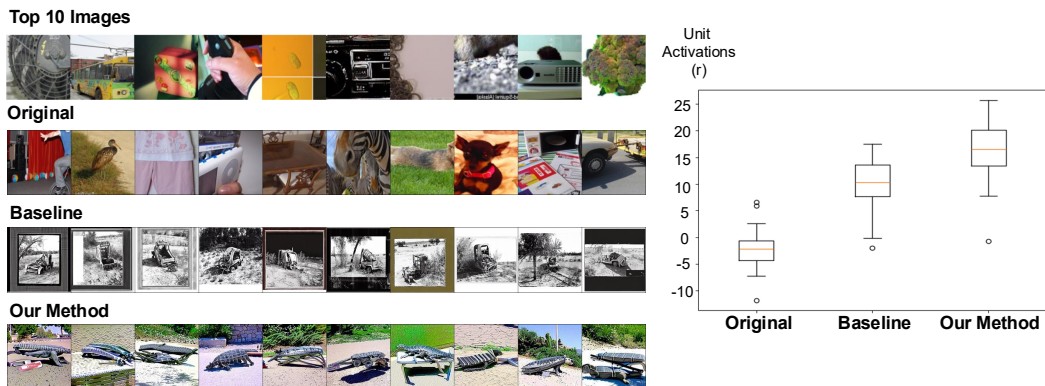

Figure S21: **One latent modification example to generate preferred images with increasing unit activations on ImageNet-mini Dataset.** Though the top 10 images can not demonstrate a notable pattern represented by the unit in the early layer of SimCLR model, our method can generate a uniform representation and compared with original images from ImageNet-mini, generated preferred images based on modifying the latent to the positive prior mean shows a more apparent change in activation compared with original images and representations from BrainACTIV.

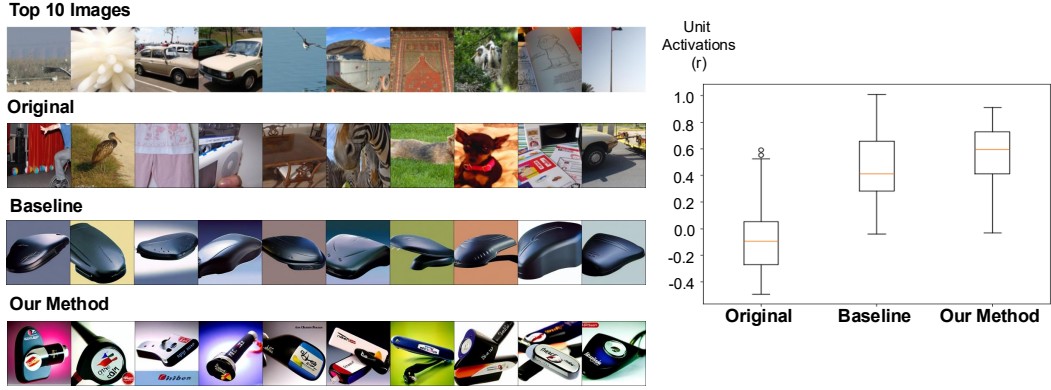

Figure S22: **One latent modification example to generate preferred images with increasing unit activations on ImageNet-mini Dataset.** Though the top 10 images can not demonstrate a notable pattern represented by the unit in the early layer of SimCLR model, our method can generate a uniform representation and compared with original images from ImageNet-mini, generated preferred images based on modifying the latent to the positive prior mean shows a more apparent change in activation compared with original images and representations from BrainACTIV.

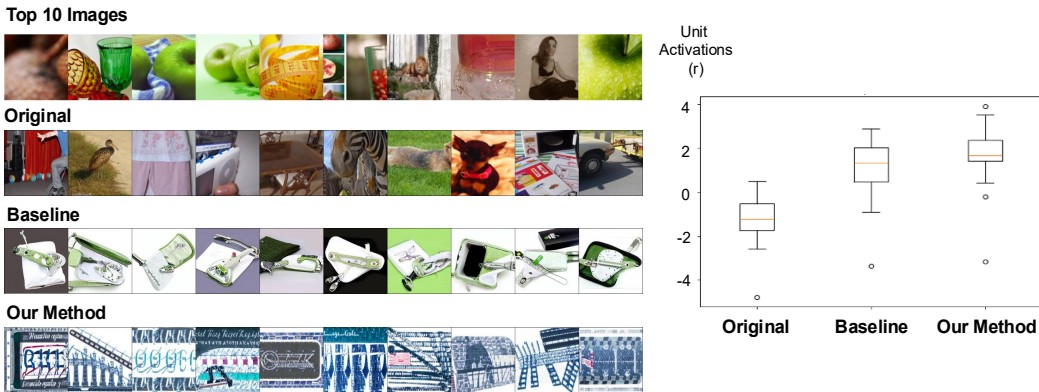

Figure S23: **One latent modification example to generate preferred images with increasing unit activations on ImageNet-mini Dataset.** Though the top 10 images can not demonstrate a notable pattern represented by the unit in the early layer of SimCLR model, our method can generate a uniform representation and compared with original images from ImageNet-mini, generated preferred images based on modifying the latent to the positive prior mean shows a more apparent change in activation compared with original images and representations from BrainACTIV.

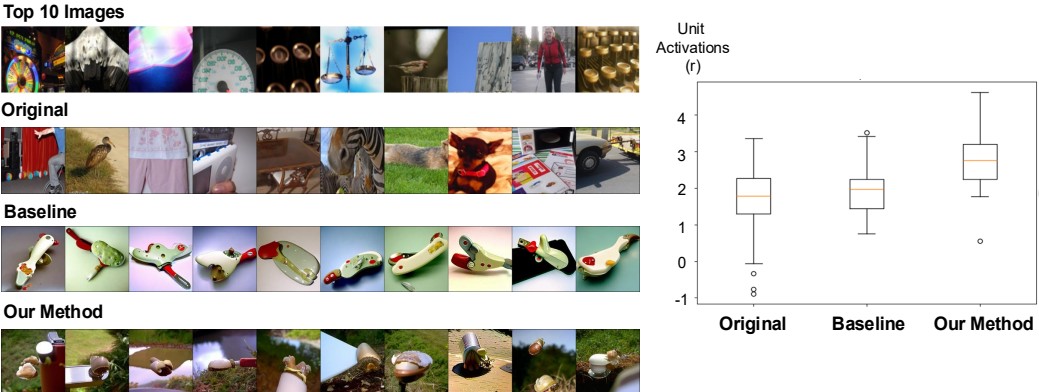

Figure S24: **One latent modification example to generate preferred images with increasing unit activations on ImageNet-mini Dataset.** The top 10 images demonstrate a notable pattern represented by the unit in the deep layer of SimCLR model, that round objects appear in top images. Our method can also generate a uniform representation and compared with original images from ImageNet-mini, generated preferred images based on modifying the latent to the positive prior mean shows a more apparent change in activation compared with original images and representations from BrainACTIV.

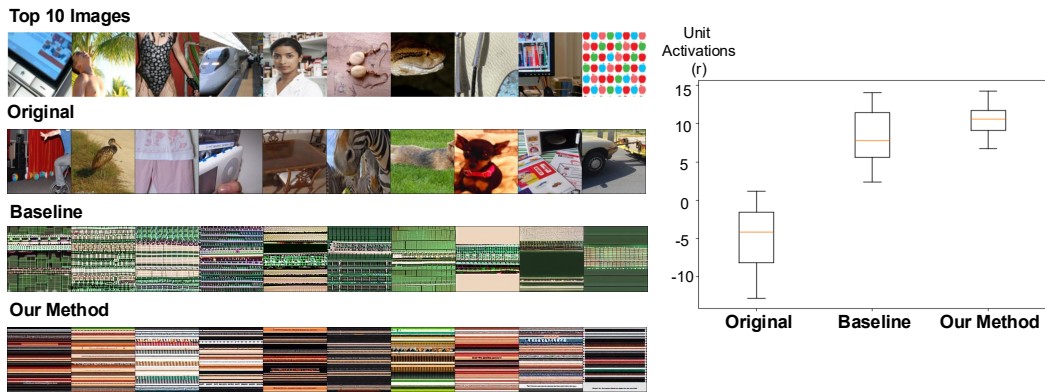

Figure S25: **One latent modification example to generate preferred images with increasing unit activations on ImageNet-mini Dataset.** The top 10 images demonstrate a notable pattern represented by the unit in the deep layer of SimCLR model, that grid objects appear in top images. Our method can also generate a uniform representation and compared with original images from ImageNet-mini, generated preferred images based on modifying the latent to the positive prior mean shows a more apparent change in activation compared with original images and representations from BrainACTIV.

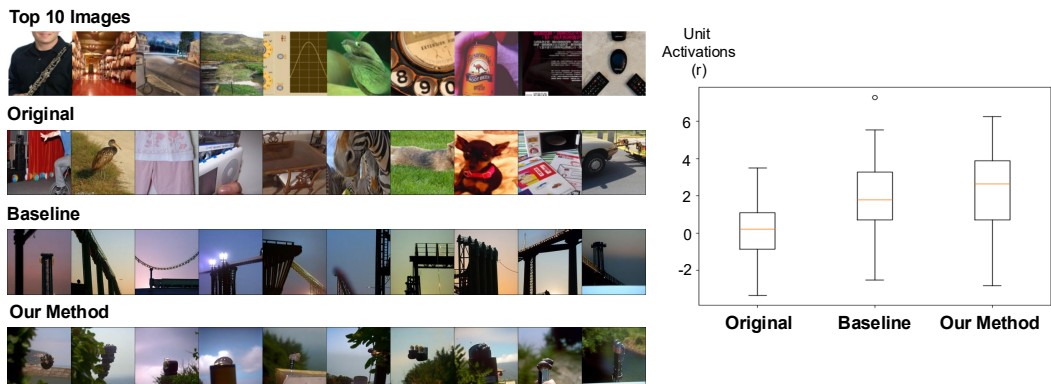

Figure S26: **One latent modification example to generate preferred images with increasing unit activations on ImageNet-mini Dataset.** Though the top 10 images can not demonstrate a notable pattern represented by the unit in the early layer of SimCLR model, our method can generate a uniform representation and compared with original images from ImageNet-mini, generated preferred images based on modifying the latent to the positive prior mean shows a more apparent change in activation compared with original images and representations from BrainACTIV.

## A.4 MORE OPTIMIZATION RESULTS

Our Optimization method based on VAE latent can be performed not only on CIFAR-10 dataset. Figure S27 and Table S2 provide additional optimization results on the ImageNet-mini dataset. Our method shows similar advantages in manipulating unit responses and preserving the features of original images.

It is worth noting that, the activation increase with our optimization method slight trails the optimization with Standard VAE, and the similarity advantage shrinks compare to CIFAR-10.

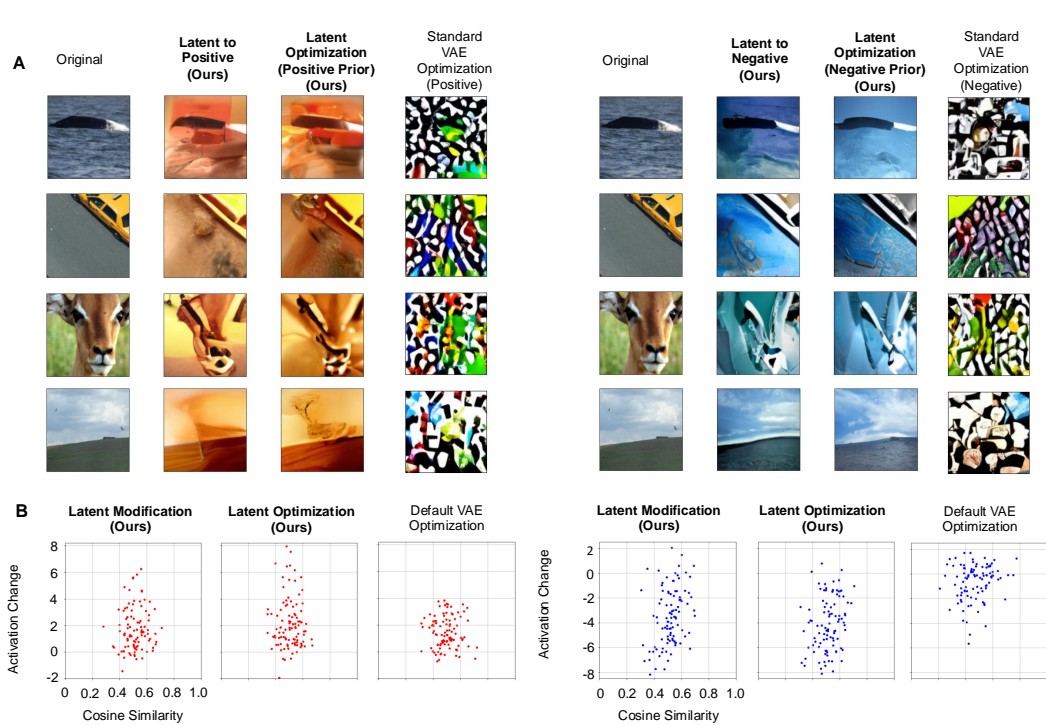

Figure S27: **a) Generating preferred images with different methods on ImageNet-mini Dataset. Left 4 columns** show optimizing images to increase their activations. **Column 1** are four example input images. **Column 2-3** are results of our method while **column 4** represent optimizing the latent space with a standard VAE. **Right 4 columns** are results of optimization in the same methods presenting in the same order, with target of decreasing activations. **b)** shows the shift in the activation value between and the cosine similarity between randomly chosen 100 original images and their optimized results over different methods. **Compared with baseline VAE**, our method can effectively increase activation better as well as maintain a greater similarity with original images, meaning more unrelated features are preserved.

Table S2: Summary of average cosine similarity and change in unit responses ($\Delta r$) between optimization results and input images over different optimization methods on ImageNet-mini dataset, corresponding to Figure S.7 B. The best results are marked with **bold face**.

| Direction | Model | Method | Average Similarity (Cosine) | Average $\Delta r$ |
|---|---|---|---|---|
| positive | structured landscape | latent modification | **0.513** | 1.710 |
| | | latent optimization | 0.503 | 2.140 |
| | original landscape | | 0.465 | **2.266** |
| negative | structured landscape | latent modification | **0.520** | -3.595 |
| | | latent optimization | 0.514 | **-4.215** |
| | original landscape | | 0.467 | -0.526 |

## B  EXPERIMENTAL SETTINGS

### B.1  DATASET DETAILS

We will show how the dataset was processed in this part.

**Natural Scenes Dataset (NSD)**:  (Allen et al., 2021): we selected 1000 images presented to Subject 1 from the dataset as our test set, while using CLIP embeddings of 9000 images shown to same subject for training. We used same test images selected by the BrainACTIV. And for the measurement of activation change and FID score, we use all 1000 images presented to same subject as test set in COCO dataset (Lin et al., 2015) and their CLIP embeddings. 2k images for FID(Heusel et al., 2017) calculation are the combination of 1k preferred images to maximize/minimize the activations, respectively. The CLIP embeddings are retrieved from the first output of CLIP encoder (Radford et al., 2021a), the dimension is 1024 for every image.

**THINGS-EEG**:  (Grootswagers et al., 2022; Gifford et al., 2022): we selected 200 images presented to Subject 1 from the dataset as our test set, while using CLIP embeddings of 16540 images shown to same subject for training. In this dataset, each image was presented four times, results in 16540 x 4 = 66160 trials. We averaged the EEG data of four trials. Then, we selected data from those time windows:

- **C1**: [0.05, 0.09](50-90ms).
- **P1**: [0.08, 0.12], Early visual processing.
- **N170**: [0.13, 0.18], Object recognition stage.
- **P2**: [0.16, 0.24], Higher-order feature processing.
- **P300**: [0.25, 0.40], Attentional evaluation.
- **LPP**: [0.40, 0.70],Late Positive Potential / Memory.

Beyond averaing trials, we also averaged all data within those time windows, to get a singular value as EEG activation data.

**Imagenet-mini** (Deng et al., 2009): The train and test split follow the default settings in the orginal dataset, which divided the whole dataset into train/val sets, we use train as our train set and val as our testset. Train set consists of 34,745 images, and test set consists of 3,923 images. For the training process of SimCLR, we used following transform techniques in torchvision.transforms to edit the orginal Imagenet-mini images:

```
transforms.RandomResizedCrop(128, scale=(0.08, 1.0),
interpolation=Image.BICUBIC)
transforms.RandomHorizontalFlip()
transforms.ToTensor()
```

**CIFAR-10** (Krizhevsky & Hinton, 2009): Also follows original settings of CIFAR-10, which are 50000 training and 10000 testing images respectively. The CIFAR-10 images in original dataset were processed with this:

```
transforms.Grayscale(num_output_channels=3)
transforms.Resize((32, 32))
transforms.ToTensor()
transforms.Normalize((0.5,), (0.5,))
```

### B.2  MODEL IMPLEMENTATION DETAILS

#### B.2.1  VAE IMPLEMENTATION AND TRAINING

**VAE:** For **NSD** and **ImageNet** dataset to learn CLIP embeddings, using VAE architecture (Kingma & Welling, 2013) consists of an encoder and a decoder with the following layers:

- **Encoder (CLIP → latent)**

- FC$_{enc}$: $1024 \longrightarrow 2048$ (first 1024 dims are $\mu$, last 1024 dims are $\log \sigma^2$)
- **Latent layer**
  - $\mu \in \mathbb{R}^{1024}, \ \log \sigma^2 \in \mathbb{R}^{1024}$
  - Reparameterisation: $z = \mu + \epsilon \odot \exp(0.5 \log \sigma^2)$
- **Decoder (latent → CLIP)**
  - FC$_{dec}$: $1024 \longrightarrow 1024$ (reconstructed CLIP embedding)

**Light-weight Bottleneck CVAE:** For **CIFAR-10** dataset, we use an image-to-image Bottleneck CVAE, the structure is:

- **Encoder**:
  - Conv1: 3→16 channels, $3 \times 3$ kernel, stride 1, padding 1, ReLU
  - Conv2: 16→32 channels, $3 \times 3$ kernel, stride 2, padding 1, ReLU
  - Conv3: 32→64 channels, $3 \times 3$ kernel, stride 2, padding 1, ReLU
- **Bottleneck**:
  - FC_mu: $64 \times 8 \times 8 \rightarrow 192$
  - FC_logvar: $64 \times 8 \times 8 \rightarrow 192$
  - FC_bn_mu: $192 \rightarrow 192$
  - FC_bn_logvar: $192 \rightarrow 192$
- **Decoder**:
  - FC_decode: $192 \rightarrow 64 \times 8 \times 8$, ReLU
  - Deconv1: 64→32 channels, $3 \times 3$ kernel, stride 2, padding 1, output padding 1, ReLU
  - Deconv2: 32→16 channels, $3 \times 3$ kernel, stride 2, padding 1, output padding 1, ReLU
  - Deconv3: 16→3 channels, $3 \times 3$ kernel, stride 1, padding 1, Sigmoid

The VAE training configurations are shown in the Table S3.

### B.2.2 Implementation of Diffusion Pipeline

All images are first converted to a 1024-dimensional representation with the frozen CLIP-ViT-H/14 encoder released by LAION (Schuhmann et al., 2022). For every split of the Natural Scenes Dataset we iterate once over the raw-pixel loader, compute the embeddings and cache the resulting CLIP features. Subsequent experiments therefore operate purely in embedding space and never revisit the expensive vision backbone.

Image synthesis uses the "img-to-img" variant of Stable Diffusion v1.5 from the DIFFUSERS library (Leocadio, 2025). We swap the original PNDM scheduler for DDIM (better speed/quality trade-off) and disable the safety checker to avoid unintended filtering. Every network in the pipeline—U-Net, text encoder, VAE, and the IP-Adapter (Ye et al., 2023) that injects the CLIP embedding into the cross-attention blocks—remains entirely frozen. We use the public 7 MB checkpoint that is aligned with the very same ViT-H/14 encoder employed for feature extraction, ensuring a loss-free conditioning path.

During inference each RGB frame ($512 \times 512$ px) is supplied together with a target CLIP embedding—either the baseline embedding obtained by ridge regression or the one produced by our activation-aware VAE. Unless noted otherwise we set the SDEdit strength to $\gamma = 1.0$; We run 50 DDIM steps per image, draw a single sample, and fix the random seed to 42 for strict comparability across all methods.

### B.2.3 Implementation of Other Models

**SimCLR and ConvNet:** For SimCLR, original ImageNet-mini was trained on 256 * 256, I re-trained the model with same configurations of SimCLR, only edited the size of input files and edited minor structures in the model to match the input size. For **CIFAR-10**: I designed a three layer ConvNet to train as the CIFAR-10 classfier, the structure is:

- **Encoder**:
  - Conv1: n_channels →32 channels, $3 \times 3$ kernel, padding 1, BatchNorm, ReLU, Max-Pool2d(2,2)
  - Conv2:32→64 channels, $3 \times 3$ kernel, padding 1, BatchNorm, ReLU, MaxPool2d(2,2)
  - Conv3:64→128 channels, $3 \times 3$ kernel, padding 1, BatchNorm, ReLU, Max-Pool2d(2,2)
- **Fully Connected Layers**:
  - FC1:128×4×4 →256 units, Dropout(0.5), ReLU
  - FC2:256 →num_classes

Table S3: VAE Hyperparameters and Training Settings for NSD, THINGS-EEG, ImageNet, and CIFAR-10 datasets

| Hyperparameter | NSD & ImageNet | THINGS-EEG | CIFAR-10 |
|---|---|---|---|
| **Data Hyperparameters** | | | |
| input_size | 1024 dim embeddings | 1024 dim embeddings | $32 \times 32$ images |
| n_channels | 1 | 1 | 3 |
| **Model Hyperparameters** | | | |
| latent_dim | 2048 | 2048 | 192 |
| **Training Settings** | | | |
| seed | 0 | 0 | 0 |
| batch_size | 128 | 128 | 128 |
| epochs | 150 | 300 | 150 |
| optimizer | Adam | Adam | Adam |
| lr | $1 \times 10^{-4}$ | $1 \times 10^{-3}$ | $1 \times 10^{-4}$ |
| scheduler | StepLR (step_size=10, $\gamma = 0.9$) | StepLR (step_size=10, $\gamma = 0.9$) | StepLR (step_size=10, $\gamma = 0.9$) |
| grad_clip | 1.0 | 1.0 | 1.0 |

## B.3 METRIC DETAILS

**Quantifying Similarity** To further evaluate the structured landscape, we used similarity and realism metrics. For similarity between original images and images with activation optimized but unrelated features preserved, we use cosine similarity as follows. Given a set of images $\{I_i\}$ with feature embeddings $\{\mathbf{f}_i\}$, the similarity score is expressed as:

$$\text{Cosine Similarity}(\mathbf{f}_i, \mathbf{f}_j) = \frac{\mathbf{f}_i \cdot \mathbf{f}_j}{\|\mathbf{f}_i\|\|\mathbf{f}_j\|}$$

**Implementation of Similarity Calculation:** For the measurement of the feature preserve capacity of different optimization methods, I used pretrained ResNet18 encoder output to represent the features on **CIFAR-10** images and Inception-V3 on **ImageNet** dataset, with consine similarity between features of original images and their corresponding output images calculated after that.

**Quantifying Realism: Frechét Inception Distance (FID):** To quantify overall realism we report the Frechét Inception Distance, computed on the 2 048-D pool-3 activations of a frozen Inception-v3 network. Let $\mu_r$, $\Sigma_r$ and $\mu_g$, $\Sigma_g$ be the empirical means and covariances of the real and generated image features, respectively. The FID is

$$\text{FID} = \|\mu_r - \mu_g\|_2^2 + \text{Tr}\big(\Sigma_r + \Sigma_g - 2\left(\Sigma_r \Sigma_g\right)^{1/2}\big).$$

Lower values indicate that the generated distribution is closer to the real one. In all experiments we follow the protocol of evaluating on 2 000 samples (1 000 originals duplicated for the real set and 1 000 synthetics per method for the generated set) and report FID computed with the TORCHMETRICS implementation under feature = 2048.

### B.4 COMPUTATIONAL RESOURCES

All experiments were carried out on a single workstation equipped with **8 × NVIDIA RTX A5000 GPUs** (each with 24 GB on-board memory, CUDA 12.2, driver 535.183), an AMD EPYC 7543 CPU (32 cores, 2.8 GHz) and 512 GB of system RAM. One GPU was used for model training or inference at a time. Training a CLIP-to-CLIP VAE on the NSD train split ($\approx 9$ k images) with a batch-size of 256 took $\approx 6.5$ hours on a single A5000; Optimizing on CIFAR-10 completed in under 40 minutes. For diffusion-based generation (IP-Adapter + Stable-Diffusion v1.5, 50 DDIM steps) ,producing one full 2 k-image evaluation set (positives + negatives) took about 70 minutes.

## C IMPLEMENTATION DETAILS

### C.1 TRAINING DETAILS

Below are the pseudo-code of training and inferrence pipeline.

---

**Algorithm 1** VAE Training with Activation Regularization

---

**Require:**
1: `vae`: Variational Autoencoder model
2: `optimizer`: Optimizer (Adam)
3: `scheduler`: Learning rate scheduler (StepLR)
4: `num_epochs`: Number of training epochs

**Ensure:**
5: VAE with structured latent space and high-quality reconstruction

6: **procedure** TRAINVAE(`vae`, `optimizer`, ...)
7:     **for** epoch = 1 to `num_epochs` **do**
8:         **for** each batch **do**
9:             `optimizer.zero_grad()`
10:             **Forward Pass:**
11:             $z \leftarrow$ `vae.encode(data)`
12:             $\hat{x} \leftarrow$ `vae.decode(z)`
13:             **Compute Loss:**
14:             Reconstruction Loss $\leftarrow \text{MSE}(\hat{x}, \text{data})$
15:             **Compute Prior Parameters:**
16:             $\sigma^{\text{prior}} \leftarrow \left| \dfrac{\lambda}{\texttt{activation\_values} + \epsilon} \right|$
17:             $\mu_i^{\text{prior}} \leftarrow \begin{cases} \mu_{\text{pos}}, & \text{if } \texttt{activation\_values}_i > 0, \\ \mu_{\text{neg}}, & \text{otherwise.} \end{cases}$
18:             KLD Loss $\leftarrow KLD_{loss}(\mu_i, \mu_i^{\text{prior}})$
19:             Total Loss $\leftarrow \omega_{\text{recon}} \times$ Reconstruction Loss $+ \omega_{\text{KLD}} \times$ KLD Loss
20:             **Backpropagation:**
21:             Total Loss.backward()
22:             `optimizer.step()`
23:         **end for**
24:         `scheduler.step()`
25:     **end for**
26: **end procedure**

---

**Latent Modification and Image Generation**    To generate high-quality preferred images that enhance activations based on the latent modification, the algorithm is shown as following block 2:

---

**Algorithm 2** Vis-Lens: Preferred–Image Generation

---

**Require:**
   $i_{\text{in}}$ ... RGB input image
   `clip_enc` ... frozen CLIP–ViT-H encoder
   $x$ ... 1024-d CLIP embedding of $i_{\text{in}}$
   `vae` ... activation–aware CLIP→CLIP VAE
   $z, \mu$ ... latent code / mean; $\mu_{\text{pos}}, \mu_{\text{neg}}$ ... activation priors
   `ip_adapter` ... frozen IP-Adapter
   `sd15` ... Stable-Diffusion v1.5 (DDIM)
   $\alpha$ ... slerp weight, $\gamma$ ... SDEdit strength
   $r_{\text{dir}} \in \{\text{pos}, \text{neg}\}$ ... desired ROI shift
**Ensure:** preferred image $i_{\text{out}}$
 1: **procedure** GENPREFERRED($i_{\text{in}}, r_{\text{dir}}$)
 2:     $x \leftarrow$ `clip_enc`($i_{\text{in}}$)
 3:     $(\mu, \log \sigma^2) \leftarrow$ `vae.encode`($x$); $z \leftarrow \mu$
 4:     **if** $r_{\text{dir}} = \text{pos}$ **then** $z' \leftarrow \text{slerp}(\alpha, z, \mu_{\text{pos}})$
 5:     **else** $z' \leftarrow \text{slerp}(\alpha, z, \mu_{\text{neg}})$
 6:     **end if**
 7:     $x_{\text{gen}} \leftarrow$ `vae.decode`($z'$)
 8:     $i_{\text{out}} \leftarrow$ `ip_adapter.generate`(clip $= x_{\text{gen}}$, image $= i_{\text{in}}, \gamma$, 50 steps)
 9:     **return** $i_{\text{out}}$
10: **end procedure**

---

Here are more detailed explanations of our generation process:

- The input image $i_{\text{in}}$ is projected to a CLIP embedding x with the frozen encoder.

- The activation-aware VAE encodes x to latent mean $\mu$; we take $z = \mu$ for a deterministic edit.

- To shift the chosen ROI, we linearly interpolate the latent toward the positive prior $\mu_{\text{pos}}$ (to raise activity) or the negative prior $\mu_{\text{neg}}$ (to suppress it), yielding z'.

- Decoding z' gives an intermediate CLIP embedding $x_{\text{gen}}$.

- $x_{\text{gen}}$ is injected—via a frozen IP-Adapter—into Stable-Diffusion v1.5. With SDEdit strength $\gamma = 1.0$ the reverse DDIM process relies entirely on $x_{\text{gen}}$; smaller $\gamma$ blends in low-level structure from the reference image.

- The sampler returns the final preferred image $i_{\text{out}}$, which is used in all downstream analyses.

## C.2 DETAILS ABOUT OPTIMIZING UNIT ACTIVATIONS WITH LATENT

The algorithm for optimizing neural network unit responses using latent variables from a VAE is listed as following algorithm block 3:

---

**Algorithm 3** Optimize Activation with Latent Variables

---

**Require:**
 1: `net`: Pre-trained neural network model
 2: `vae`: Pre-trained Variational Autoencoder
 3: `layer_name`: Target layer in `net`
 4: `unit_idx`: Index of target unit
 5: `num_iterations`: Number of optimization steps
 6: `learning_rate`: Learning rate for optimizer
 7: `KLD_weight`: Weight for Kullback-Leibler Divergence loss
 8: `target`: Direction "max" or "min")
 9: `init_data`: Initial data sample

**Ensure:**
10: Optimized responses scores, Controlled KLD loss

11: **procedure** OPTIMIZEACTIVATION(`net`, `vae`, . . .)
12:     **Initialize Latent Variable**:
13:     $(\mu, \text{logvar}) \leftarrow$ `vae.encode(init_data)`
14:     $z \leftarrow \mu$
15:     **Set Up Optimizer**:
16:     `optimizer` $\leftarrow$ Adam($z$, lr=`learning_rate`)
17:     **Register Activation Hook** on `layer_name`
18:     **for** iteration = 1 to `num_iterations` **do**
19:         **Decode Latent to Image**:
20:         $x \leftarrow$ `vae.decode`($z$)
21:         **Forward Pass**:
22:         `net(x)`
23:         **Compute Loss**:
24:         **if** `target` = "max" **then**
25:             loss $\leftarrow -$activation $+ \omega_{\text{KLD}} \times \text{KLD}(z)$
26:         **else**
27:             loss $\leftarrow$ activation $+ \omega_{\text{KLD}} \times \text{KLD}(z)$
28:         **end if**
29:         **Backpropagation**:
30:         optimizer.zero_grad()
31:         loss.backward()
32:         optimizer.step()
33:     **end for**
34:     **Remove Activation Hook**
35:     **Return** $x$, activation scores
36: **end procedure**

---

**More Explanation of the Algorithm**   In this algorithm, we aim to optimize the activation of specific units in a neural network model by adjusting the latent variables $z$ of a pre-trained Variational Autoencoder (VAE). The primary goal is to maximize or minimize the target unit's activation while preserving the overall image quality.

**Initialization:** We initialize the latent variable $z$ by encoding the input image using the VAE encoder:

$$(\mu, \text{logvar}) \leftarrow \texttt{vae.encode(init\_data)}$$
$$z \leftarrow \mu$$

**Optimization Loop:** For each iteration, we perform the following steps:

1. **Decode Latent to Image:** The current latent variable $z$ is decoded to generate an image $x$:

$$x \leftarrow \texttt{vae.decode}(z)$$

2. **Forward Pass and Activation Capture:** The generated image $x$ is passed through the neural network `net`. We utilize a registered **hook function** on the target layer to capture the activation of the target unit during this forward pass:

$$\texttt{net}(x)$$

The hook function retrieves the target activation activation, which will be used in the loss computation. This approach allows us to monitor and manipulate the activation of specific units without altering the forward pass logic of the network.

3. **Compute Loss:** We compute the loss function, which consists of two components: the activation term and the Kullback-Leibler Divergence (KLD) regularization term.

- **Activation Term:** Depending on the optimization direction (maximize or minimize), we use the negative or positive of the target unit's activation:

$$\text{loss} = \begin{cases} -\text{activation}, & \text{if } \texttt{target} = \text{``max''} \\ \text{activation}, & \text{if } \texttt{target} = \text{``min''} \end{cases}$$

- **KLD Regularization Term:** The KLD term ensures that the latent variable $z$ remains within a plausible region of the latent space.

We consider two method for the KLD calculation. One is **our method**, and the other one is using the latent space of **Standard VAE** to compute KLD loss, which is our **baseline**.

- **Our Method:** We calculate the KLD using a modified prior that incorporates the desired change in activation. Specifically, the prior mean $\mu^{\text{prior}}$ is adjusted based on the target direction:

$$\mu^{\text{prior}} = \begin{cases} +1, & \text{if } \texttt{target} = \text{``max''} \\ -1, & \text{if } \texttt{target} = \text{``min''} \end{cases}$$

The KLD is then computed as:

$$\begin{aligned} \text{KLD}_{\text{our}}(z) = \frac{1}{2} \sum_{i=1}^{d_z} & \left[ \log\left( \frac{\sigma_i^{\text{prior}}}{\sigma_i} \right) - 1 \right. \\ & \left. + \frac{\sigma_i^2 + (\mu_i - \mu_i^{\text{prior}})^2}{(\sigma_i^{\text{prior}})^2} \right] \end{aligned}$$

where $\sigma_i^{\text{prior}}$ is set to a small constant (e.g., 0.1).

- **Standard VAE:** We calculate the KLD using the standard Gaussian prior with zero mean and unit variance:

$$\text{KLD}_{\text{standard}}(z) = \frac{1}{2} \sum_{i=1}^{d_z} \left( \mu_i^2 + \sigma_i^2 - \log(\sigma_i^2) - 1 \right)$$

- **Total Objective Function:** The total loss combines the activation loss and the KLD regularization:

$$\text{loss} = \text{activation\_loss} + \omega_{\text{KLD}} \times \text{KLD}(z)$$

where $\text{KLD}(z)$ is either $\text{KLD}_{\text{our}}(z)$ or $\text{KLD}_{\text{standard}}(z)$ depending on the mode.

4. **Backpropagation:** We perform backpropagation to compute the gradients of the loss with respect to $z$ and update $z$ using the optimizer.

$$\text{optimizer.zero\_grad}()$$

loss.backward()

optimizer.step()

**Post-processing with DDPM:** To enhance the quality and realism of the generated image, we further refine $x$ using a pre-trained Denoising Diffusion Probabilistic Model (DDPM).

**Final Results:** The final optimized images presented in our paper are the outputs from the DDPM refinement process, not the direct outputs from the VAE optimization.

**Configurations:** During the implementation, learning rate was set to 0.1, num_iterations to 10000 and $\omega_{\text{KLD}}$ to 0.001.

## D    PREDICTIVE POWER OF IN-SILICO BRAIN SIMULATOR

Here are the graphs of the predictive power of our In-silico brain simulator. To improve the training efficiency, the In-silico brain simulators are training with different ROI groups.

**ROIs:** OFA, FFA-1, EBA, VWFA-1, OPA, PPA, RSC, OWFA:

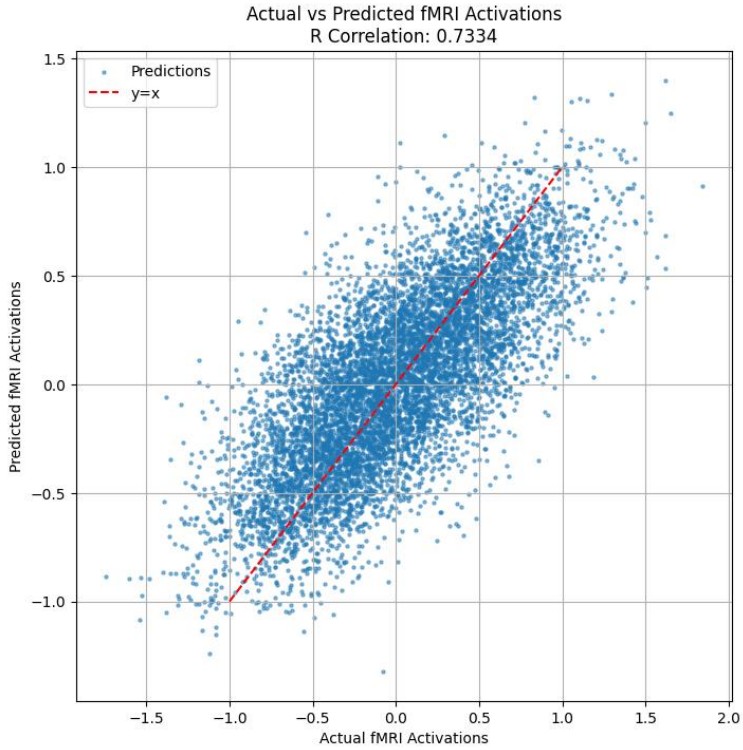

**ROIs:** IPS4, SPL1, TO1, PHC2, VO2, PGi, VVC, TE2a:

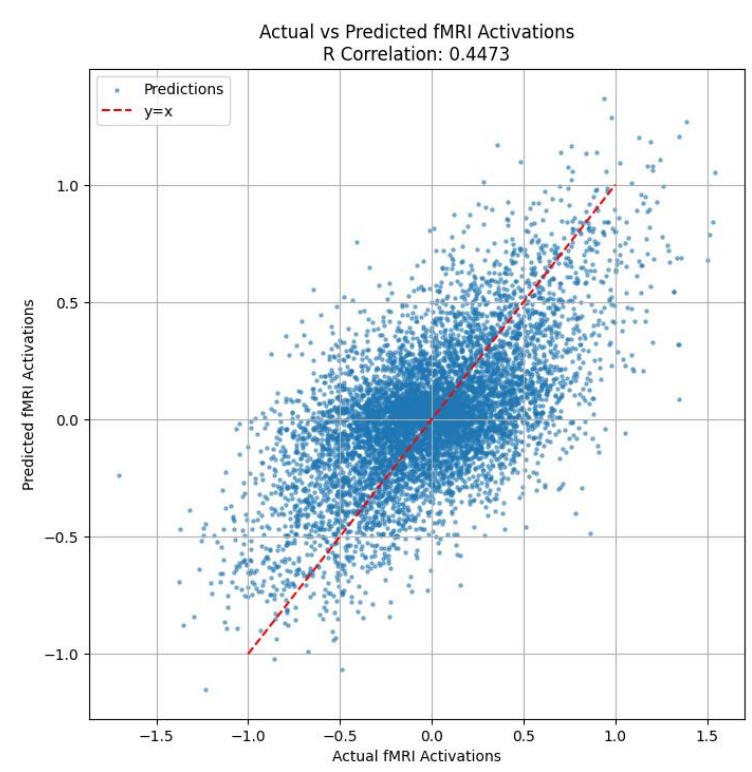

**ROIs:** STSdp, 31a, 31pd, DVT, FST, Gp, PGs, PH:

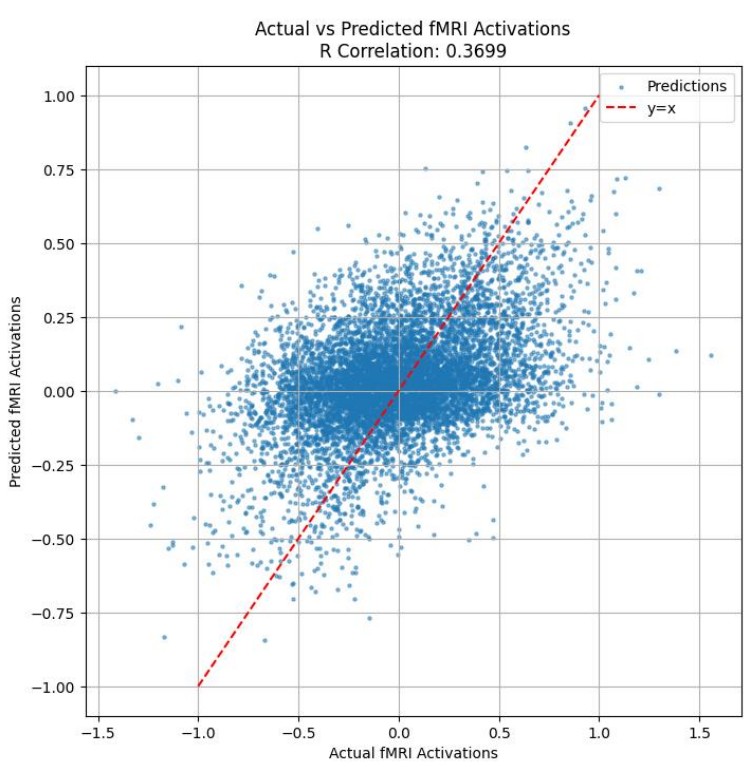

**ROIs:** TE2p, PHT, TPOJ1, TPOJ2, TPOJ3, IP0, IP1, IP2:

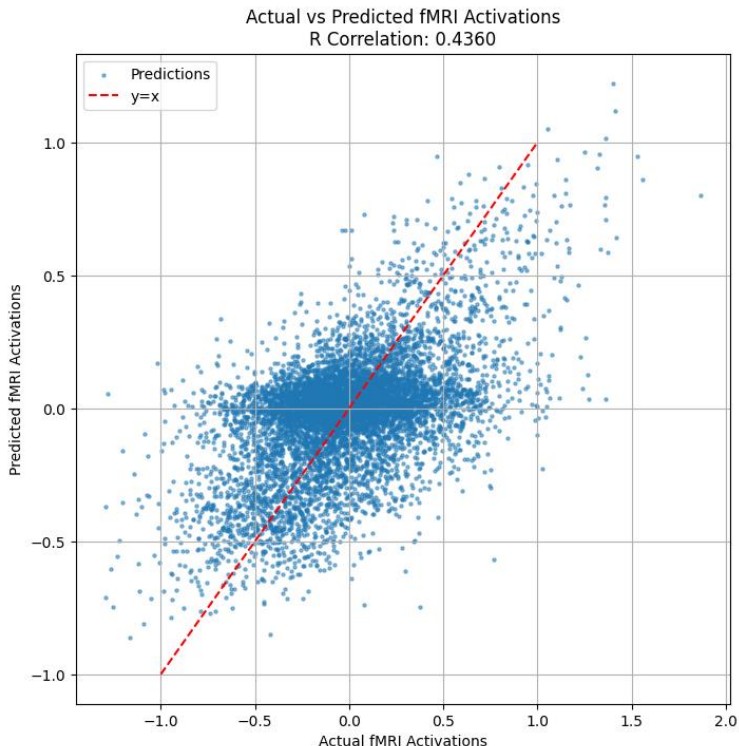

**ROIs:** V1, V2, V3, V4, V8, LO1, LO2, V4t :

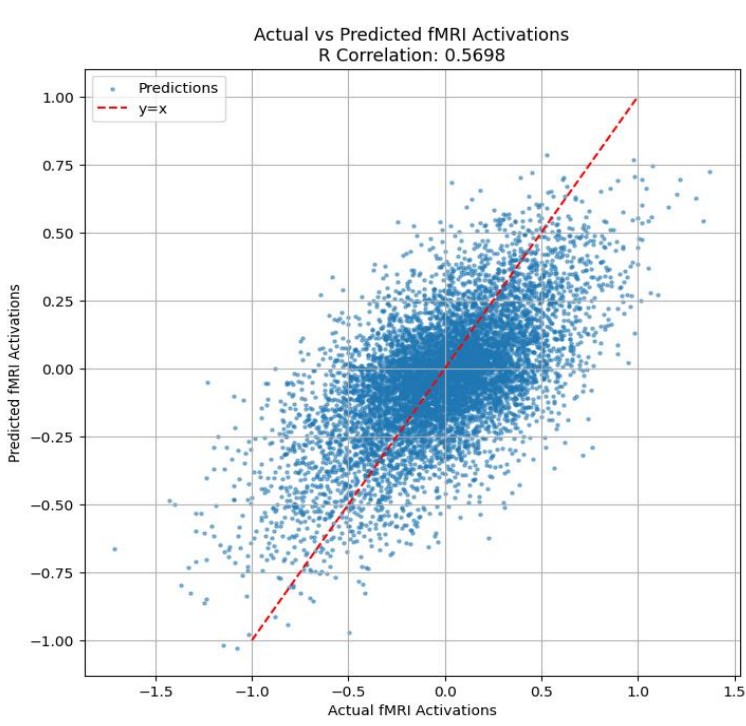

**Occipital Channels:** O1, Oz, O2, PO7, PO3, POz, PO4, PO8:

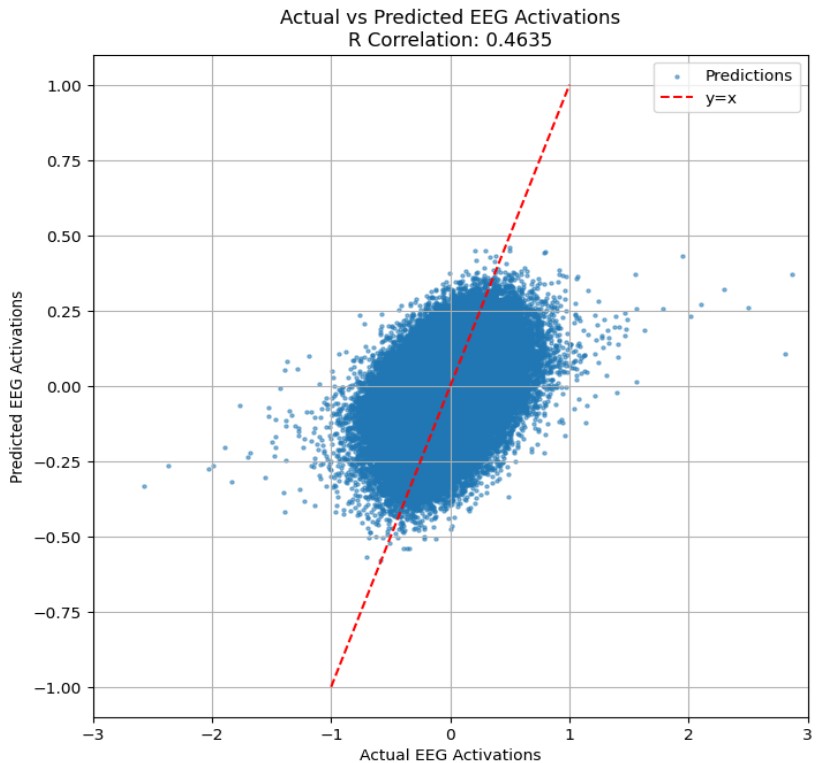

