# OpenReview forum: "Structured Visual Landscape: Generating Preferred Representations in Multi-modal Biological and Artificial Neural Networks"
_ICLR.cc/2026/Conference — Submitted to ICLR 2026_

### Official Review · Reviewer_ZZhw · 2025-10-30

**Soundness:** 2
**Presentation:** 4
**Contribution:** 1
**Rating:** 4
**Confidence:** 4

**Summary:**

The authors introduce VisLens which uses activation-regularized VAEs to generate images that drive or suppress activations in artifitial and biological NNs. This method learns a structured latent space that supports the generation of realistic and controllable stimuli without iterative optimization. The evaluation uses the NSD fMRI and Things-EEG datasets and different networks. Results show that compared to baseline methods such as BrainACTIV the proposed method generates images that better modulate activations, provides better cross-subject generalization and produces more realistic images.

**Strengths:**

I was able to easily read and follow the paper, it flows well and I found no typography errors or awkward phrasing.

The results were properly tested on a wide variety of datasets, including EEG, fMRI, and multiple neural network activations.

The authors committed to releasing all the models and data

Cross subject generalization: This important aspect is rarely tested. I was happy to see the authors thoroughly examine this.

**Weaknesses:**

Limited novelty, see contribution and soundness sections

The FID is a somewhat contrived metric. The distribution of the generated images might match the distribution of the original data more closely pixel wise. However, to examine that it is indeed more realistic a small human preference survey might be necessary.

Previous papers such as "driving and suppressing the human language network" recorded brain activity for the generated stimuli that is believed to maximally drive / suppress different ROIs. This might be excessive, but is an idea that might be worth exploring, especially for thingsEEG.

The methods seem to be sound as far as the proposed framework working. However, I believe the improvements in FID scores and other results are mainly due to the constraints of the regularized VAE inducing structure to the latent space that "stabilize" sampling during image generation. In other words, to generate images vectors are sampled from a latent space that was constrained to have a linear shape, and due to this constrain the risk of the data points being "out-of-distribution" (leaving the manifold of natural images in the latent space) is minimized, resulting in more coherent / realistic images. In other words, the improved performance seems to be due to better sampling during image generation, with the cost being that we have to learn a transformation into a more structured latent space. It is unclear to me that this method captures anything about the data that BrainACTIV does not, or that the results can not be beat by more sophisticated sampling in BrainACTIV pipeline.

The contribution as far as I can tell is that the authors learned an extra transformation into more structured space, allowing for better sampling during generation. It is unclear to me that this method captures anything about the data that BrainACTIV does not, or that the results can not be beat by more sophisticated sampling in BrainACTIV pipeline. More analysis would be needed in this direction.

**Questions:**

I would suggest randomly picking N images generated by BrainActiv and the proposed method and examining if humans actually see any significant difference between them.

Previous papers such as "driving and suppressing the human language network" recorded brain activity for the generated stimuli that is believed to maximally drive / suppress different ROIs. This might be excessive, but is an idea that might be worth exploring, especially for thingsEEG.

---

> ### Author Response · Authors · 2025-12-03
>
> Thanks for your constructive and encouraging comment! I’m more than glad to try my best to address your concerns one by one.
>
> **Human evaluations:**  We agree with the reviewer that human evaluation is an excellent standard to measure realism, but conducting new human subject experiments (fMRI/EEG) is beyond the scope of this conference paper, which focuses on establishing a computational framework.
>
> **Illustration on “better samplings”:** We appreciate this insightful comment. The reviewer is correct that our method stabilizes image generation by constraining latents to make sampling "in-distribution". However, we respectfully disagree that this is merely a sampling trick or that the baseline could achieve the same results simply through more sophisticated sampling. The difference is fundamental to how the neural signal is modeled:
>
> 1. **Non-Linearity vs. Linear Vectors:** The BrainACTIV baseline assumes that the relationship between CLIP embeddings and brain activity is linear ($r = w \cdot x + b$). It defines the "preferred" image solely by moving along a single direction vector $w$. No amount of sophisticated sampling can correct this if the underlying neural representation is non-linear or sparse. In contrast, our VAE learns a non-linear mapping, allowing it to capture complex, mixed-selectivity features that a single linear vector cannot represent.
> 2. **Asymmetry of Activation:** As shown in our results, the baseline often fails to decrease activation because it assumes the "minimum" is simply the mathematical reverse of the "maximum" ($z_{min} = -w$). Neural representations are rarely perfectly symmetric in semantic space. Our method captures this by learning distinct priors for positive and negative activations, which is why we significantly outperform the baseline in activation decrease tasks.
> 3. **Signal Uncertainty (The Variance Constraint):** Our method captures something specific about the data that BrainACTIV ignores: **uncertainty**. By scaling the prior variance inversely with activation strength ($\sigma^{prior} = |\frac{\lambda}{r+\epsilon}|$), Vis-Lens naturally filters out noisy, low-activation data points, whereas a linear model tries to fit them. This allows our sampler to be "guided" by the strength of the signal, not just its direction.
>
> Therefore, the "cost" of learning the transformation yields a structured landscape that handles the non-linear, asymmetric, and noisy nature of biological data while a linear baseline cannot address regardless of how it samples. We will revise the manuscript in the future work to clarify these theoretical distinctions.

---

### Official Review · Reviewer_4H6N · 2025-10-31

**Soundness:** 2
**Presentation:** 1
**Contribution:** 2
**Rating:** 2
**Confidence:** 5

**Summary:**

This paper proposes a novel two-stage method to generate minimally/maximally activating images for functional selectivity visualization of units or regions in biological (fMRI, EEG) and artificial neural networks. In the first stage, a VAE is trained to reconstruct CLIP embeddings while being regularized to a neural activation prior, creating a structured latent space where embeddings are organized from minimally to maximally activating. In the second stage, a reference image is encoded in this latent space, and interpolated there towards min or max activation; the interpolated embedding is then passed through the decoder, and the output, now in CLIP space, is used to guide the manipulation of the reference image with a diffusion model SDEdit. Compared to a recent prior baseline that does not have this structured latent space, the proposed method gives better results for minimizing brain activations, as well as for maximizing activations in some brain areas with mixed selectivity. Authors claim that the method is comparable to the baseline in the highly selective regions, and that it outperforms it when minimizing/maximizing activations in ANNs.

**Strengths:**

The paper offers a novel method that improves performance of prior methods in certain respects (better activation minimization, maximization for mixed-selectivity), although not across the board. It is methodologically solid, and its advantage in minimizing activations can be clearly linked to the method, as stated in “our method produces greater decrease in activation in all regions, demonstrating effectiveness of defining an independent negative prior when constructing landscape rather than simply reverse the maximal direction in baseline”. Finally, the method is explained in a fairly clear manner, and is compared against recent state-of-the-art methods.
However, presentation quality is quite poor, and some of the claims are not adequately supported with evidence.

**Weaknesses:**

Listed from more major to more minor:
1. Results (section 4.3.1) do not support the claim that the proposed method *generally* outperforms the Garcia et al. method used as baseline. Specifically, results across all ROIs (including those in the Appendix) show that it outperforms the baseline in the “Activation Decrease” but not consistently in the “Activation Increase”. This benefit for the “Decrease” is by itself an important contribution and should be presented as the main contribution of the paper.

    For the “Activation Increase”, it is claimed that the method outperforms the baseline in mix-selectivity regions like IPS4, SPL1, 31a, TE2p, IP0, and IP1 while it is comparable in highly selective regions like FFA, OFA, EBA, and OPA. However, in OFA, FFA, and EBA, the method actually *underperforms* the baseline, if we are to judge the score difference by the same standards as in the other regions. Out of all the regions (in the main paper), it seems the ones the two methods do perform comparably on are TE2p, IP0, and OPA (2 mixed selectivity and 1 high), although it is hard to tell without statistical significance. A more accurate account would be that the method outperforms the baseline in 4/6 mixed-selectivity regions, and underperforms the baseline in 3/4 high-selectivity regions. Thus, the argument that the method is beneficial for “Activation Increase” in mixed-selectivity regions could potentially also stand as a secondary contribution of the paper, but it needs to also extend to the (high-level) ROIs in the Appendix. The disadvantage of the method for highly selective regions should also be made clear as a limitation.

    Finally, the usefulness of generating maximally or minimally activating images for specific EEG electrodes (which do not have good spatial resolution and cannot be claimed to exclusively correspond to specific brain regions) is debatable. Also, EEG results do not show clear advantage against the baseline in decreasing or increasing activations.
2. Similarly the evidence does not support the claim well enough in the results concerning preferred representations of ANNs instead of brain (section 4.3.2), where the improvement shown is not convincing at all with missing statistical tests, and the differences (esp. in B) looking very marginal. It is also unclear how the specific ANN units are chosen; would it not be better to show results across many units? Also, could this analysis not have been performed with the same images as the ones in NSD and THINGS instead of switching to ImageNet?
3. No statistical significance is computed for the score differences between proposed method and baseline, making it hard to discern between meaningful differences and marginal ones. Also, no standard deviation is reported in Table 1, even though it exists in Figure 3B as vertical bars. The origin of the deviation shown in those bars is also not explained. Are they across all reference images from the test set? Also in Figure 4 no statistical test is performed between the box plots. Additionally, the reason for picking the specific ROIs for the main paper while leaving the rest to the Appendix should be better motivated.
4. Results section 4.3.3 is overall hard to understand. Specifically, what is the “certain unit” that controls the latent space activated by and how is it chosen? Why is CIFAR used, which has really tiny and hard to interpret images? It would be good to also show how similar to the original are the images generated in this setting by the baseline method of Garcia et al.
5. In the paper title and in many instances throughout the paper the word “Multimodal” is mentioned - it is not clear at all what this refers to. The datasets NSD and THINGS-EEG are not multimodal (they show visual stimuli to the subjects, multimodal would be if they showed other modalities of stimuli like multimodal video with sound, or text-image pairs).  If it refers to having both fMRI and EEG, the word “multimodal” is not commonly used to refer to this and is misleading, could be replaced with “multiple neural brain imaging methods” or similar for clarity. Same goes if “multimodal” refers to dealing with both neural data and images, the word would be misleading. If it refers to the embedding space coming from CLIP, this does not seem enough to justify such focus on multimodality, as only the visual encoder is used. Thus, it is suggested that the word “Multimodal” be removed.
6. There are many issues in the Presentation quality of the paper.
    - The writing (grammar, tenses, style) is very poor with numerous errors throughout the paper, which would be impossible to outline all of them here. In order to match the quality of the venue it needs to be read and corrected by a native or fluent English speaker, perhaps a colleague of the authors.
    - The flow of ideas does not always make sense and some things are mentioned out of nowhere (i.e. do not follow from prior text), examples: “which avoids the tedious optimization procedure” (abstract - what is this procedure?), “brain computer interface applications” (line 106, 485 - like what?), “avoiding falling into local minimum in the optimization process” (line 161 - not mentioned as a problem of the baseline before), “pass it to a frozen SDEdit diffusion model” (line 312 - this model is not introduced in the section above, and the gamma parameter not defined).
    - Contextualization relative to prior work is too historical (not relevant enough) in the first two paragraphs, it is not clear how GANs are related or the early work in feature visualization except from a purely historical perspective. In the Introduction with the current order of paragraphs 2 and 3, it is not made clear that the methods cited in paragraph 2 are also used for the same goal as the proposed method in paragraph 3.
    - There are various problems with the paper figures. In Figure 2 it looks like posterior=prior in the top box, which is not true. Labels are missing from x, y, and z axes of the latent space. SDEdit is used but not explicitly shown in the method figure. Figure 3B does not show absolute scores clearly enough, they are very hard to discern with no ticks or intermediate values on the y axis. Figure 4 is missing titles for the figures on the right (should be Sim-CLR and ViT for top and bottom respectively).

**Questions:**

For the paper to be considered for acceptance (rating of 6 maximum), all of the above weaknesses need to be addressed with a concrete change in the manuscript.

List of additional questions for clarification:
- How are the two latent dimensions shown in Figure 1 obtained? Are they principal components of the latent space?
- How is the scalar activation r computed? Is it a scalar across all voxels inside an ROI? Additionally, is it also averaged across stimuli?
- Were absolute EEG amplitudes considered for r? As a (highly) negative response is also a high activation in EEG.
- How are the prior means mu_pos and mu_neg “predefined”? How are they set?
- In Table 2, is the held-out subject only considered for a single fold, or is it averaged across multiple folds of held-out subjects (cross-validated)?

---

> ### Author Response · Authors · 2025-12-03
>
> We thank the reviewer for their thorough evaluation and constructive feedback. We value the suggestions regarding our work like the framing of our contributions, the statistical significance of our results, and the presentation of the manuscript. We are committed to addressing these points in the future work as detailed below.
>
> **Refining contribution claims.** Follow your suggestions, we will revise the text in our future work to present "Activation Decrease" as our primary contribution. Regarding "Activation Increase," we will explicitly distinguish between our superior performance in mixed-selectivity regions (e.g., IPS4, SPL1) versus our comparable performance in highly selective regions (e.g., FFA).
>
> **EEG and ANN experiments.** We would focus on presenting the results about the high-temporal-resolution components in EEG. For ANNs, we will expand our analysis to include a broader range of randomly selected units. The reason why we choose ImageNet is to test our method on a popular and more diverse image dataset, in order to demonstrate generalizability. We will include the intuition for this decision in the future work.
>
> **Statistical analysis.** We will update related tables and figures to include more statistical analysis results like Confidence Intervals(CI) for all results and comparisons, to provide stronger support for our claims.
>
> **Clarification on section 4.3.3.** The "certain unit" is a unit in the specific convolutional layers and CIFAR-10 is an example used to demonstrate our method. Our major focus here is to develop a new optimization method with proposed structured landscape. Also, the main goal is to compare optimization performances across different VAE latent spaces(Structured vs Original), so the “baseline” here should be the VAE with a default latent space, rather than the method of Garcia et al, which does not involve any VAE latent space.
>
> **Improve presentation.** We sincerely apologize for any confusions in our manuscript and thank the reviewer for pointing them out. We will replace the ambiguous term with more precise terminology. We will also undertake a comprehensive proofread to fix grammatical errors, improve logical flow, and correct mistakes in visual components.
>
> Also, we are happy to answer following questions one by one here:
>
> **Derivation of latent dimensions.** The latent dimensions shown in Figure 1 are the first and second dimension in VAE latent space, they are not principal components.
>
> **Computation of scalar activation r.** $r$ is the mean value of all voxels in a region, it is computed by averaging the fMRI/EEG data across all voxels given a specific region/time window, and averaged across stimuli.
>
> **Consideration of absolute EEG amplitudes.** Yes, our method considers absolute EEG amplitudes as we define a different cluster center for negative responses, and they will also be treated as a strong constraint to construct structured latent space according to VAE prior settings.
>
> **Definition and sensitivity of prior means.** $\mu_{pos}$ and $\mu_{neg}$ are two separate values serving as the means of VAE prior. We use an arbitrary setting where we set them to 1/-1 for $\mu_{pos}$/$\mu_{neg}$. It’s worth noting that the performances are not sensitive to these values during the experiments, for example, on area OFA, changing $\mu$ to -2/2 results in a negligible change for activation manipulation, compared to $\mu = -1/1$.
>
> **Evaluation protocol for held-out subjects.** It is evaluated on a fixed test set which is split by the NSD dataset, so the held-out-subject evaluation only includes a single fold.

---

### Official Review · Reviewer_dETa · 2025-11-01

**Soundness:** 2
**Presentation:** 3
**Contribution:** 2
**Rating:** 2
**Confidence:** 4

**Summary:**

This paper proposes a method for linking visual stimuli to neural activity by learning structured latent spaces that can generate images predicted to elicit specific activation patterns in biological and artificial neural networks. The approach integrates generative modeling and neural encoding to explore how visual features map onto activity within different regions of interest. The innovation lies in regularizing both high- and low-activity spaces, aiming to capture the full spectrum of meaningful neural variation. While the conceptual framework is interesting, the empirical validation is limited, with several aspects of the evaluation lacking quantification and clarity.

**Strengths:**

The paper addresses an important problem in computational neuroscience and interpretable AI—how to generate and analyze visual stimuli that meaningfully modulate neural activity. Its main innovation is to learn a latent space that is explicitly regularized for both high and low neural activity, capturing the full spectrum of neural tuning rather than focusing solely on activation increases. The inclusion of both increased and decreased activation directions is novel as far as I know and aligns with recent findings that low-activity patterns can also carry significant information. The technical implementation, combining a variational autoencoder with a diffusion model, is appropriate for the stated goals. The use of CLIP embeddings and pre-trained diffusion models provides a solid and practical foundation, even if not particularly innovative.

**Weaknesses:**

- Despite the above strengths, the paper falls short in quantitative validation and clarity. The authors claim that their method helps to alleviate the effects of noise in neural data, but it remains unclear why this would be the case, and empirically it is not clear whether it actually is.


- The paper does not show that the approach is more robust for particularly noisy data compared to data with higher signal-to-noise ratios, nor does it quantify how the proposed regularization contributes to stability. Beyond this, several technical aspects of the evaluation require clarification.


- In Table 1, the reported activation changes are not accompanied by statistical measures, making it difficult to determine whether the improvements exceed inter-subject variability. The qualitative neuron examples in Figure 4 are interesting but insufficient to assess generality across the population.


- The discussion section is very short and would benefit from additional depth, especially regarding the implications of the findings and how they relate to prior methods. The related work section on feature visualization also misses some recent and relevant studies, for example Fel et al. (2023) [1] which would help situate the contribution more clearly within current literature.


- Another limitation is the lack of clarity on how much data is required to train the latent space for each region or neuron. If this is done on a per-neuron basis, the method may become data-intensive or impractical for broader application. The supplementary figures, while extensive, do not provide a clear sense of which method performs better or why. In addition, the generated images appear to have systematically higher contrast, which could itself drive neural responses; it is unclear whether contrast or similar low-level image properties were controlled for during the analysis.


[1] https://arxiv.org/abs/2306.06805

**Questions:**

1. Can the authors provide quantitative evidence that their method is more robust to noise in neural data, and clarify why the proposed regularization should lead to such robustness?


2. In Table 1, can they include measures of variability to demonstrate that the reported activation changes exceed inter-subject variance?


3. Regarding Figure 4, how consistent are the neuron-level effects across the population, and are the differences shown in the plots statistically significant? It would also be helpful to clarify how low-level image properties, such as contrast, were controlled for, given their strong influence on neural activity.


4. Finally, how much data is required to train the latent space, and does the data requirement scale with the number of neurons or regions of interest? Including predicted activations—for example using the DINOv2 encoding model—for the supplementary examples would also make the comparison between methods more interpretable. (edited)

---

> ### Author Response · Authors · 2025-12-03
>
> We thank the reviewer for the insightful comments and for acknowledging the innovation of our activation-regularized latent space. We appreciate the opportunity to clarify the robustness of our method, the statistical validation, and technical details regarding data requirements.
>
> **Robustness to noise.** Our method demonstrates more robustness in that it can generate visual representations that can regulate the neuronal activations to desired directions on all regions. This is because, compared with fitting all the data linearly, we imposed a soft constraint on the neuronal recordings, that scales the variance of the latent prior inversely with the activation magnitude; this allows the model to treat weaker, potentially noisy signals with higher uncertainty while strictly aligning with strong neural responses.
>
> **Variability and inter-subject variance in Table 1.** We are happy to provide the activation change results along with variability on the NSD dataset to prove that the reported activation change can exceed the inter-subject variance. The comparison between the original predictor and Subject 2 is as follows:
>
> | **ROIS**                                  | **OFA**       | **FFA-1**     | **EBA**       | **VWFA-1**    | **OPA**       | **PPA**       | **RSC**       | **OWFA**      |
> | ----------------------------------------- | ------------- | ------------- | ------------- | ------------- | ------------- | ------------- | ------------- | ------------- |
> | Activation Increase on original predictor | 0.869± 0.117  | 0.851± 0.128  | 0.860± 0.136  | 0.867± 0.113  | 0.731± 0.128  | 0.966± 0.143  | 0.942± 0.140  | 0.567± 0.157  |
> | Activation Increase on subj 2             | 0.861± 0.121  | 0.975± 0.139  | 0.790± 0.080  | 0.648± 0.120  | 0.629± 0.160  | 1.061± 0.204  | 0.940± 0.202  | 0.462± 0.155  |
> | Activation Decrease on original predictor | -0.526± 0.118 | -0.382± 0.113 | -0.564± 0.116 | -0.279± 0.103 | -0.557± 0.151 | -0.610± 0.148 | -0.775± 0.158 | -0.592± 0.151 |
> | Activation Decrease on subj 2             | -0.436± 0.075 | -0.302± 0.134 | -0.391± 0.112 | -0.089± 0.114 | -0.485± 0.152 | -0.813± 0.191 | -0.823± 0.212 | -0.566± 0.177 |
>
> **Consistency of neuron-level effects and control for low-level properties.** Due to the constraint of our computational resources, we are unable to cover all neurons within ANN, even for a single layer in a pretrained large vision encoder, whose embedding size is usually in the thousands. We will include the statistical analysis in related paragraphs in our future work. In Figure 4, activations are retrieved on late layers in the vision encoder, where some high-level concepts are expected to be found by our method. We are glad to include the results of some early layers in the vision encoder in the future, as we may learn some low-level image properties from there.
>
> **Data requirements for training.** We included the data settings in Appendix B.1; around 9000 images are used to train the latent space. We did not see a different requirement of dataset scale on different ROIs. Those settings align well with the baseline method such that the results are comparable.

---

### Official Review · Reviewer_QtiM · 2025-11-04

**Soundness:** 2
**Presentation:** 2
**Contribution:** 3
**Rating:** 2
**Confidence:** 4

**Summary:**

This paper introduces Vis-Lens, a novel framework for generating preferred visual representations in biological and artificial neural networks (ANN). The core contribution is the proposed activation-regularized prior for Variational Auto-Encoder (VAE), leverages activations r from a target brain region or ANN unit to organize the VAE's latent space. By sampling from this structured space and passing the modified latent embeddings to a frozen diffusion model, Vis-Lens can generate new images that effectively modulate target activations without requiring per-image optimization. Experiments on human brain data and image data demonstrate model effectiveness on brain and artificial neural networks.

**Strengths:**

-	A new, simple and efficient prior for VAE to learn structured visual latent landscape, conditioned on the brain activation.
-	The proposed method shows advantageous performance compared to baseline, on multi-modal human brain data (fMRI, EEG) and image data in both activation modulation and image realism (FID).
-	Potentials for neuroscience analysis and brain visual understanding.

**Weaknesses:**

1) The contributions seem limited.  The author does not provide intuition, explanation or theoretical support to justify why the proposed prior is effective. Why \sigma_{prior} is defined as Eq2?
2) Several technical details remain unclear:
2.1)What is activation r? Does that value come from brain signal data like fMRI and EEG? What does it represent? How is it used for ANN?
2.2)What are \mu_{pos} and \mu_{neg}?
3)What are metrics Activation Increase and Decrease? What is its meaning and how is it calculated?
4)The discussion of prior work seems limited; there have been several VAE and related machine learning approaches for similar problem ("Generative decoding of visual stimuli." In International Conference on Machine Learning, pp. 24775-24784. PMLR, 2023; "Hierarchical VAEs provide a normative account of motion processing in the primate brain." Advances in Neural Information Processing Systems 36 (2023))
5) Ablation studies (Table 3) show that using structured landscape with latent modification (main method) obtain lower \delta_r than standard VAE (0.046 vs 0.067), raising concern about method effectiveness. What are the results on brain data?
6) Lacking parameter studies. How hyperparameters such as \lambda, \alpha, are selected and how they contribute to the performance?
7) How training/inference time of Vis-Lens compared to baseline?

**Questions:**

1) The contributions seem limited.  The author does not provide intuition, explanation or theoretical support to justify why the proposed prior is effective. Why \sigma_{prior} is defined as Eq2?
2) Several technical details remain unclear:
2.1)What is activation r? Does that value come from brain signal data like fMRI and EEG? What does it represent? How is it used for ANN?
2.2)What are \mu_{pos} and \mu_{neg}?
3)What are metrics Activation Increase and Decrease? What is its meaning and how is it calculated?
4)The discussion of prior work seems limited; there have been several VAE and related machine learning approaches for similar problem ("Generative decoding of visual stimuli." In International Conference on Machine Learning, pp. 24775-24784. PMLR, 2023; "Hierarchical VAEs provide a normative account of motion processing in the primate brain." Advances in Neural Information Processing Systems 36 (2023))
5) Ablation studies (Table 3) show that using structured landscape with latent modification (main method) obtain lower \delta_r than standard VAE (0.046 vs 0.067), raising concern about method effectiveness. What are the results on brain data?
6) Lacking parameter studies. How hyperparameters such as \lambda, \alpha, are selected and how they contribute to the performance?
7) How training/inference time of Vis-Lens compared to baseline?

---

> ### Author Response · Authors · 2025-12-03
>
> We thank the reviewer for their detailed feedback and for recognizing the efficiency and potential of our Vis-Lens framework. We appreciate the opportunity to clarify the theoretical basis of our prior, technical definitions, and experimental results. Please find our responses below.
>
> **Theoretical justification for the prior formulation.** Rather than only involving a random “new prior” for VAE, we have several reasons to define such a prior. Firstly, defining a variance ($1/|r|$) that decreases as the absolute value of neuronal response ($r$) increases, which indicates a strong positive/negative activation, added with well-separated mean values ($\mu_{pos}$ & $\mu_{neg}$) relying on the sign of the activation, allows us to construct a structured VAE latent space encouraging the images with maximized/minimized neuronal activations to be mapped to separable centers, while loosening such constraints when the activations is relatively weak. Meanwhile, by defining this prior in VAE, we successfully developed a method based on probabilistic formulas, which is different from the previous work in this field.
>
> **Definition of activation $r$.** Generally, activation $r$ is the value of neuronal response. The actual meaning of $r$ depends on the source of neuronal data. When the value comes from brain signal data (fMRI), it is a value derived from the original recordings from the dataset (recordings are averaged across specific regions or time windows). For ANN, $r$ is retrieved from the output of middle layers of pretrained ANNs, by selecting units from those layers and collecting their corresponding output values.
>
> **Role of $\mu_{pos}$ and $\mu_{neg}$.** These are two mean values of the VAE prior that we defined; in the Vanilla VAE with a gaussian prior, it is usually defined as singular value 0. Also, they represent two centers in VAE latent space where the inputs with maximized/minimized $r$ are located theoretically.
>
> **Metrics for Activation Increase/Decrease.** These metrics represent the predicted activation differences between the generated preferred images and original images, by editing the latent code of VAE to positive/negative priors ($\mu_{pos}$ and $\mu_{neg}$).
>
> **Inclusion of prior work.** We sincerely apologize for the limited discussion and will include the suggested references in the future work.
>
> **Clarification on Table 3 comparisons.** This is a good observation, however, there are some misinterpretations of the table. The 0.046 (latent modification with structured landscape) and 0.067 (gradient descent on the original landscape directly) are not comparable, for those are two different methods. The correct comparison is made between gradient descent methods on structured and original landscapes to demonstrate the effectiveness of our proposed structured landscape, and our landscape can better increase the activation (0.079 vs 0.067) while keeping more features of the original image (0.666 vs 0.520).
>
> **Results on brain data.** For Sec 4.3.3, its main goal is to demonstrate the strength of our structured VAE latent space compared with original space, instead of verifying it on various datasets, so we did not include brain data in the paper, but we are happy to provide the brain data results in the future work.
>
> **Selection and impact of $\alpha$.** For the $\alpha$ setting, it does not affect the conclusion of the paper due to our goal being to compare the representations learned by our method and baseline, so setting $\alpha$ to 1.0 is sufficient to do such comparison. Meanwhile, we presented the visual changes in images by setting different alphas (Interpolation power), and showed them in Figure 3 and Appendix Figure S1-S20.
>
> **Selection and impact of $\lambda$.** For $\lambda$, it does have a major effect on the final outcome, and we have chosen to report the results under best $\lambda$ settings, and we are willing to include results on different $\lambda$ settings in future work.
>
> **Comparison of training and inference time.** The baseline training time takes about 20 mins while ours takes about 40 mins on our device. Inference time is equivalent for both methods because the VAE reconstruction (our method) and the linear interpolation (baseline) are almost instantaneous.

---

### Meta-Review · Area_Chair_sTfh · 2026-01-03

**Summary:**

The paper proposes an improved regularization method for generating images maximizing or minimizing activations in biological or artificial neural networks. The method is rvaluated on fMRI and EEG data, as well as artificial networks, and shows promising results.

The reviewers agree that while the paper is interesting and has its merit, it is not fit for publication at the moment. The main issues are:
- Limited and not always statistically significant improvement over the baseline. Especially for activation maximization
- Somewhat limited contribution compared to existing methods.

While the rebuttal addressed some of the questions, I do not think it addressed these core concerns. I therefore recommend rejection at this point. I believe the paper might be more appropriate for a specialized workshop.

**Reviewer Concerns:**

The rebuttal addressed some questions about technical details (meaning of different variables and equations) and details of experiments and results.

I do not think it really resolved the concerns about the limited improvement over the baseline and limited contribution.

**Reviewer Scores:**

I think one or two of the reviewers might have increased their scores by 1 point, but this would not change the final decision.

---

### Decision · Program_Chairs · 2026-01-26

Reject